# Protein undernutrition reduces the efficacy of praziquantel in a murine model of *Schistosoma mansoni* infection

**Joseph Bertin Kadji Fassi**[1,2], **Hermine Boukeng Jatsa**[1,2]*, **Ulrich Membe Femoe**[1,2,3], **Valentin Greigert**[3], **Julie Brunet**[3], **Catherine Cannet**[4], **Christian Mérimé Kenfack**[1,2], **Nestor Gipwe Feussom**[1,2], **Emilienne Tienga Nkondo**[1,2], **Ahmed Abou-Bacar**[3], **Alexander Wilhelm Pfaff**[3], **René Kamgang**[1], **Pierre Kamtchouing**[1], **Louis-Albert Tchuem Tchuenté**[2,5]

1 Laboratory of Animal Physiology, Department of Animal Biology and Physiology, Faculty of Science, University of Yaoundé I, Yaoundé, Cameroon, 2 Centre for Schistosomiasis and Parasitology, Yaoundé, Cameroon, 3 Institute of Parasitology and Tropical Diseases, Dynamic Host-Pathogen Interactions, University of Strasbourg, Strasbourg, France, 4 Laboratory of Histomorphometry, Institute of Legal Medicine, University of Strasbourg, Strasbourg, France, 5 Laboratory of Parasitology and Ecology, Department of Animal Biology and Physiology, Faculty of Science, University of Yaoundé I, Yaoundé, Cameroon

* mjatsa@yahoo.fr

**Data Availability Statement:** All relevant data are within the manuscript.

## Abstract

### Background

Undernutrition and schistosomiasis are public health problems and often occur in low and middle-income countries. Protein undernutrition can alter the host-parasite environment system and aggravate the course of schistosomiasis. This study aimed to assess the impact of a low-protein diet on the efficacy of praziquantel.

### Methodology/Principal findings

Thirty-day-old mice were fed with a low-protein diet, and 40 days later, they were individually infected with fifty *Schistosoma mansoni* cercariae. A 28-day-treatment with praziquantel at 100 mg/kg for five consecutive days followed by distilled water begins on the 36th day post-infection. Mice were sacrificed on the 64th day post-infection. We determined the parasitological burden, liver and intestine histomorphometry, liver injury, and immunomodulation parameters. Praziquantel treatment of infected mice fed with a standard diet (IN-PZQ) resulted in a significant reduction of worm and egg burdens and a normalization of iron and calcium levels. The therapy also improved schistosomiasis-induced hepatopathy and oxidative stress. The anti-inflammatory and immunomodulatory activities of praziquantel were also significant in these mice. When infected mice receiving the low-protein diet were treated with praziquantel (ILP-PZQ), the body weight loss and hepatomegaly were not alleviated, and the worm and liver egg burdens were significantly higher than those of IN-PZQ mice ($P < 0.001$). The treatment did not reduce the increased activities of ALT and γ-GGT, the high malondialdehyde concentration, and the liver granuloma volume. The iron and calcium levels were not ameliorated and differed from those of IN-PZQ mice ($P < 0.001$ and $P < 0.05$). Moreover, in these mice, praziquantel treatment did not reverse the high level of IL-

**Funding:** One of the authors of this work (UMF) received the financial support of the France Government through the Cooperative and Cultural Action Service (SCAC) scholarship 959139G and the Strasbourg Institute of Parasitology and Tropical Diseases (IPPTS) to perform cytokine and RT-qPCR analyses. The funders had no role in study design, data collection and analysis, decision to publish, or preparation of the manuscript.

**Competing interests:** The authors declare that they have no competing interests.

5 and the low mRNA expression of CCL3/MIP-1α and CXCL-10/IP-10 induced by *S. mansoni* infection.

## Conclusion/Significance

These results demonstrated that a low-protein diet reduced the schistosomicidal, antioxidant, anti-inflammatory, and immunomodulatory activities of praziquantel.

### Author summary

Almost 90% of people requiring schistosomiasis preventive chemotherapy in 2018 lived in sub-Saharan Africa. Besides, 205.3 million children under five years suffer and die of undernutrition in low- and middle-income countries. The physiopathology of schistosomiasis mansoni involves liver damage, oxidative stress, and perturbation of the immune response. These disturbances are intensified by undernutrition. Praziquantel is used to treat schistosomiasis, but its efficacy on the comorbidity of *S. mansoni* infection and undernutrition has not been investigated. We conducted this study to assess the effectiveness of praziquantel on *S. mansoni* infection in mice fed with a low-protein diet. We recorded growth retardation, hepatomegaly, and high worm and egg burdens in mice fed with a low-protein diet and treated with PZQ. Moreover, the treatment did not reverse the liver function injury, oxidative stress, high iron level, and low calcium level. The proinflammatory cytokine IL-5 was still high, and the gene expression of some macrophage-associated chemokines was reduced. Therefore, this study demonstrated that in a murine model of a low-protein diet, the efficacy of praziquantel on *S. mansoni* infection was reduced. It also underlines the importance of targeting protein deficiency and malnutrition in populations living in schistosomiasis endemic areas for efficient disease control.

## Introduction

Undernutrition and schistosomiasis are public health problems and often share the same geographical areas. Approximately 700 million persons, primarily children, are at risk of schistosomiasis in 78 endemic countries. It is estimated that more than 229 million people worldwide, with almost 90% of them living in sub-Saharan Africa, required preventive chemotherapy in 2018. [1]. Estimates by the World Health Organization show that 462 million adults are underweight, 205.3 million children under five years suffer from undernutrition, and about 45% of deaths are linked to undernutrition and mainly occur in low- and middle-income countries [2]. *Schistosoma mansoni* infection induces liver damage through the granulomatous inflammatory formation around eggs trapped in the sinusoidal periportal spaces and the generation of reactive oxygen species (ROS) [3–5]. Undernourished people are generally more susceptible to infections and increased morbidity and mortality [6–10]. Protein malnutrition induces structural changes in the lymphoid organs and impairs the innate and adaptative immune response. It is thus recognized as the cause of frequent immunodeficiency [11–15].

The relationship between protein malnutrition of the host and *S. mansoni* infection is a very complex mechanism, not entirely determined since each can increase the other. Protein malnutrition is a factor that can alter the host-parasite environment system, aggravating the course of schistosomiasis by breaking the equilibrium in the relationship among the components of this system [16–20]. Clinical and experimental studies have demonstrated the interference of malnutrition in the outcome of schistosomiasis or vice versa. Assis et al. [17,21]

showed that *S. mansoni* infection negatively impacts schoolchildren's growth at low or moderate levels. The association between inadequate dietary intake and heavy *S. mansoni* infection increased the risk of stunting children. Other authors indicated that the association between undernutrition and *S. mansoni* infection leads to growth retardation, intensifies liver injuries, and decreases the humoral immune response in mice [19,22,23]. Moreover, feeding mice dams with a restricted-protein diet lead firstly to neonatal malnutrition of offspring during lactation and secondly to an increased egg output and liver damage in *S. mansoni*-infected pups [20]. More specifically, morphometric studies revealed that undernutrition of the host impairs the somatic development, the reproductive system, and the tegumental structure of male adult *S. mansoni* worms recovered from undernourished infected mice [18]. In clinical studies, praziquantel treatment improved the anthropometric indexes, nutritional status, and haemoglobin level of *S. mansoni* or *S. japonicum*-infected children and adolescents [17,24]. However, the literature lacks data on the effect of praziquantel on the comorbidity of *S. mansoni* infection and undernutrition. This study aimed to assess the efficacy of praziquantel on *S. mansoni* infection in mice fed with a low-protein diet.

## Materials and methods

### Ethics statement

All procedures in this study followed the principles of laboratory animal use and care of the "European Community" guidelines (EEC Directive 2010/63/EEC) and were approved by the "Animal Ethical Committee" of the Laboratory of Animal Physiology of the Faculty of Sciences, University of Yaoundé I–Cameroon.

### Animals

Animals used for this study were bred in the "Centre for Schistosomiasis and parasitology" of Yaoundé, Cameroon. Thirty-day-old BALB/c mice were separated from their mother and weighed. They were housed in polypropylene cages with free access to food and water and maintained under natural 12 h light/ 12 h dark cycles at temperatures between 22 and 25°C. *S. mansoni* intermediate host *Biomphalaria pfeifferi* snails were collected from the river "Afeme" (Yaoundé, Cameroon) and maintained in the laboratory under standardized conditions.

### Experimental protocol

We separated 30 days-old mice into two pools during this study and fed them two different diets: a standard diet and a low-protein diet. The bromatological characteristics of these diets are shown in Table 1.

**Table 1. Bromatological characteristics of the diets used for mice feeding.**

| Nutrient content | Standard diet | Low-protein diet |
|---|---|---|
| Protein (%) | 22.05 | 14.60 |
| Fat (%) | 3.08 | 3.52 |
| Crude fiber (%) | 7.08 | 7.17 |
| Calcium (%) | 2.00 | 1.46 |
| Phosphorus (%) | 1.43 | 1.17 |
| Sodium (%) | 0.37 | 0.30 |
| Lysine (%) | 1.42 | 0.77 |
| Methionine (%) | 0.50 | 0.32 |
| Metabolizable energy (kcal) | 2292.78 | 2333.94 |

**Table 2. Experimental design.**

| Type of diet | Mice Groups | Status of infection | Treatment | Number of mice |
|---|---|---|---|---|
| Standard diet | HN | Not infected | Distilled water | 6 |
| | IN | *S. mansoni*-infected | Distilled water | 9 |
| | IN-PZQ | *S. mansoni*-infected | PZQ | 9 |
| Low-protein diet | HLP | Not infected | Distilled water | 6 |
| | ILP | *S. mansoni*-infected | Distilled water | 9 |
| | ILP-PZQ | *S. mansoni*-infected | PZQ | 9 |

HN: healthy mice receiving a standard diet; IN: infected-untreated mice receiving a standard diet; IN-PZQ: infected mice receiving a standard diet and treated with praziquantel; HLP: healthy mice receiving a low-protein diet; ILP: infected-untreated mice receiving a low-protein diet; ILP-PZQ: infected mice receiving a low-protein diet and treated with praziquantel

We induced protein malnutrition in 30 days-old mice by feeding them a low-protein diet to mimic the usual diet of people with low-income revenue in endemic areas of schistosomiasis [25–27].

After forty days of feeding, we divided mice into six different groups, as mentioned in Table 2. Mice belonging to the infection groups were individually infected with 50 *S. mansoni* cercariae by the tail and legs immersion technique [28] and then left for 35 days for schistosome maturation and mating.

Treatment started at day 36 post-infection, and PZQ was administered *per os* at the dose of 100 mg/kg for five consecutive days, followed by a daily administration of 10 mL/kg of distilled water. Praziquantel tablets (Merck KGaA, Darmstadt, Germany) were ground into powder and dissolved in distilled water. The duration of the treatment was 28 days, and mice were sacrificed by cervical dislocation at day 64 post-infection [29,30].

## Measurement of the body and organ weights

Each mouse was weighed once a week during the experiment to assess the body weight variation from the day of *S. mansoni* infestation until the end of praziquantel treatment.

We determined the Lee index to appreciate the growth of animals based on their nutritional status. We measured the body lengths (nose-anal) the day before the sacrifice (day 63 post-infection) for all groups and calculated the Lee index using the following formula:

Lee index = cube root of body weight (g) / nose-to-anus length (cm) [31]

After mice sacrifice, their liver, spleen, and intestine were removed and weighed individually. The relative weight (g of organ/100 g of body weight) or the organ weight index was then calculated.

## Determination of worm and egg burdens

On the 64th day post-infection, we recovered adult *S. mansoni* worms from the porto-mesenteric system of the liver by perfusion and counted them under a stereo-microscope [32]. The percentage of reduction of worm burden was calculated as follows:

$$P = [(C - V)/C] \times 100$$

Where P = percentage of reduction; C = mean number of worms recovered from *S. mansoni*-infected and untreated mice; V = mean number of worms recovered from *S. mansoni*-infected and treated mice;

*S. mansoni* ova were counted in the feces the day before the sacrifice (63rd day post-infection). Feces were individually collected from each infected mouse, weighed, and homogenized in 10% buffered formaldehyde. Two aliquots of 100 μL each were counted on a light microscope to determine the number of eggs. After sacrifice, the mice's liver and intestine were removed, rinsed with PBS, weighed, and digested separately in 4% KOH solution at 37°C for 6h. Tissue suspensions were centrifuged at 1500 rpm for 5 min, and the supernatant was removed [33]. Using a light microscope, the number of eggs was determined in two aliquots of 100 μL each using a light microscope. Results were expressed as the mean number of eggs per gram of feces or tissue for the liver and intestine.

## Evaluation of biochemical biomarkers of the liver function

We collected blood from the retro-orbital venous plexus in EDTA and dry tubes. We used blood collected in EDTA tubes for hematology analysis using an automated hemacytometer (Sysmex XN-1000).

Blood collected in dry tubes was centrifuged at 3500 rpm for 15 min, and the serum obtained was stored at -70°C for biochemical analysis. Therefore, we determined the total protein level using the Biuret method [34] and the albumin level using the BIOLABO kit according to the method described by Doumas et al. [35]. In addition, we measured the activity of alanine aminotransferase (ALT) and aspartate aminotransferase (AST) according to the Reitman and Frankel method by using Bioclin kits [36]. Furthermore, we estimated the activities of alkaline phosphatase (ALP) and Gamma-glutamyltransferase (GGT) as described by Burtis et al. [37] and Szasz et al. [38], respectively. We also determined the total bilirubin, glucose, iron, calcium, total cholesterol, and triglycerides according to the protocol described in the BIOLABO kits by Burtis et al. [37]. In addition, the protocol to assay HDL cholesterol was described by Badimon et al. [39]. As a result, LDL cholesterol level was calculated as follows:

[LDL cholesterol] = [total cholesterol]–[HDL cholesterol]–([triglycerides] / 5)

## Evaluation of oxidative stress biomarkers in the liver

After mice sacrifice, we homogenized 0.4g of the liver in 2 mL of a 50 mM Tris-HCL buffer, pH 7.4, with a mortar on ice. The homogenate (20% w/v) was centrifuged at 3000 rpm for 25min at 4°C. The supernatant was collected and stored at -70°C until assayed. We then evaluated lipid peroxidation by measuring malondialdehyde (MDA) as described by Wilbur et al. [40]. Furthermore, we measured catalase and superoxide dismutase activities following the protocol described by Sinha [41] and Misra et Fridovish [42]. In addition, we determined reduced glutathione (GSH) and nitrite concentrations using Ellman's reagent [43] and the protocol described by Hibbs et al. [44], respectively.

## Determination of the serum levels of cytokines

We measured cytokine levels in serum samples using the mouse ProtocartaPlex 10-plex immunoassay kit (Thermofisher Scientific, Vienna, Austria), following the manufacturer's recommendations. Data were acquired on the Luminex XMAP technology (Merck Millipore, Darmstadt, Germany) using the Luminex XPONENT software solutions. The concentration (pg/mL) of each cytokine was determined by interpolating the median fluorescent intensity (MFI) of a dilution standard curve over seven dilution points supplied with the kit and calculated by the ProtocartaPlex Analyst 1.0 software (eBioscience, Thermo Fisher Scientific, USA). The immune mediators were classified into four categories: T helper-1 cytokines (IL-2, IFN-γ, and TNF-α), T helper-2 cytokines (IL-4, IL-5, and IL-13), T helper-17 cytokine (IL-17A), and regulatory cytokines (TGF-β1 and IL-10).

**Table 3. Mice primers used for RT-qPCR.**

| mRNA | Forward Primer (5' 3') → | Reverse primer (5' 3') → | Amplicon size |
|---|---|---|---|
| GAPDH | AGGTCGGTGTGAACGGATTTG | TGTAGACCATGTAGTTGAGGTCA | 123 |
| CCL2/MCP-1 | TTAAAAACCTGGATCGGAACCAA | GCATTAGCTTCAGATTTACGGGT | 121 |
| CXCL-10/IP10 | CCAAGTGCTGCCGTCATTTTC | TCCCTATGGCCCTCATTCTCA | 133 |
| CCL3/MIP-1α | TTCTCTGTACCATGACACTCTGC | CGTGGAATCTTCCGGCTGTAG | 100 |
| IFN-γ | ACAGCAAGGCGAAAAAGGATG | TGGTGGACCACTCGGATGA | 106 |
| IL-10 | GCTCTTACTGACTGGCATGAG | CGCAGCTCTAGGAGCATGTG | 105 |
| IL-13 | CCTGGCTCTTGCTTGCCTT | GGTCTTGTGTGATGTTGCTCA | 116 |
| FGF1 | CCCTGACCGAGAGGTTCAAC | GTCCCTTGTCCCATCCACG | 122 |
| TGF-β1 | CTCCCGTGGCTTCTAGTGC | GCCTTAGTTTGGACAGGATCTG | 133 |
| FoxP3 | CCCATCCCCAGGAGTCTTG | ACCATGACTAGGGGCACTGTA | 183 |

## Real-time quantitative polymerase chain reaction analysis (RT-qPCR)

We extracted total RNA from 10 mg of liver tissue using a NucleoZOL reagent (Macherey-Nagel GmbH & Co. KG, Germany). First, we determined RNA concentration using a Nano-Drop 2000c spectrophotometer (Thermo Scientific, USA), and each RNA sample was reversely transcribed into cDNA (cDNA Synthesis kit, Quanta Biosciences, Inc, USA). Next, we mixed the obtained cDNA with the Universal Supermix kit (Bio-Rad, USA). After that, we amplified it using CFX Real-Time System (Bio-Rad, USA), with glyceraldehyde 3-phosphate dehydrogenase (GAPDH) as a reference gene. Finally, we calculated the expression of each cDNA using a standard curve and expressed the relative expression of mRNA in each sample as a GAPDH ratio. Primers used for RT-qPCR were purchased from Eurofins Genomics (Nantes, France) and are shown in Table 3.

## Morphological and morphometry studies of liver and intestine

After perfusion of mice for worm recovery, the liver and intestine were removed, rinsed with PBS, weighed, and divided into portions. Next, we fixed pieces of the liver (left, square, and caudate lobes) and the intestine (ileum) in 10% formaldehyde prepared in PBS. After trimming and dehydration in ethanol baths, we embedded tissues in paraffin and 5 μm thick sections cut using a microtome. A set of these sections was stained with hematoxylin-eosin (HE) to evaluate the granulomatous inflammatory response. Another set was stained with picrosirius-red (PS) to evaluate fibrosis through collagen deposition in the liver or the intestine. Images were acquired at x10 and x20 using CMOS Digital Camera (IDS, Germany) connected to an optical microscope (Axiophot, Germany) and analyzed by PathScan Touch software (Excilone, France).

We carried out a morphometry analysis of liver and intestine sections stained with HE. We analyzed images captured with a digital camera AmScope MD 500 (AmScope, USA) connected to an optical microscope with ImageJ 1.32 software (NIH, USA). The following parameters were evaluated: the number of liver and intestine granulomas and the volume of the liver granuloma. In addition, we estimated the number of granulomas per microscopic field and calculated the volume of each granuloma using the formula of the sphere volume ($4/3\pi R^3$) [30,45].

## Statistical analysis

Data are expressed as means ± SEM. Data were analyzed using GraphPad Prism version 8.01 for Windows by one-way analysis of variance (ANOVA), and differences between groups were

assessed using Tukey's multiple comparison post-test. Differences were considered significant at $P < 0.05$.

## Results

### Praziquantel treatment does not improve the body weight and the liver and intestine weight indexes of *Schistosoma mansoni*-infected mice receiving a low-protein diet

As shown in Fig 1, healthy mice receiving either a standard diet or a low-protein diet gained weight during the experimentation. On the contrary, infected mice (IN and ILP groups) significantly lost weight compared to their respective healthy controls HN and HLP ($P < 0.001$) from the tenth week of experimentation (4 weeks *p.i*) to the end (10 weeks *p.i*). Oral administration of praziquantel to infected mice receiving the standard diet induced a normalization of body weight as the body weight variation increased in the IN-PZQ group compared to IN group ($P < 0.01$). On the contrary, the body weight of infected mice receiving the low-protein diet and treated with PZQ (ILP-PZQ group) was not improved after the treatment. The body

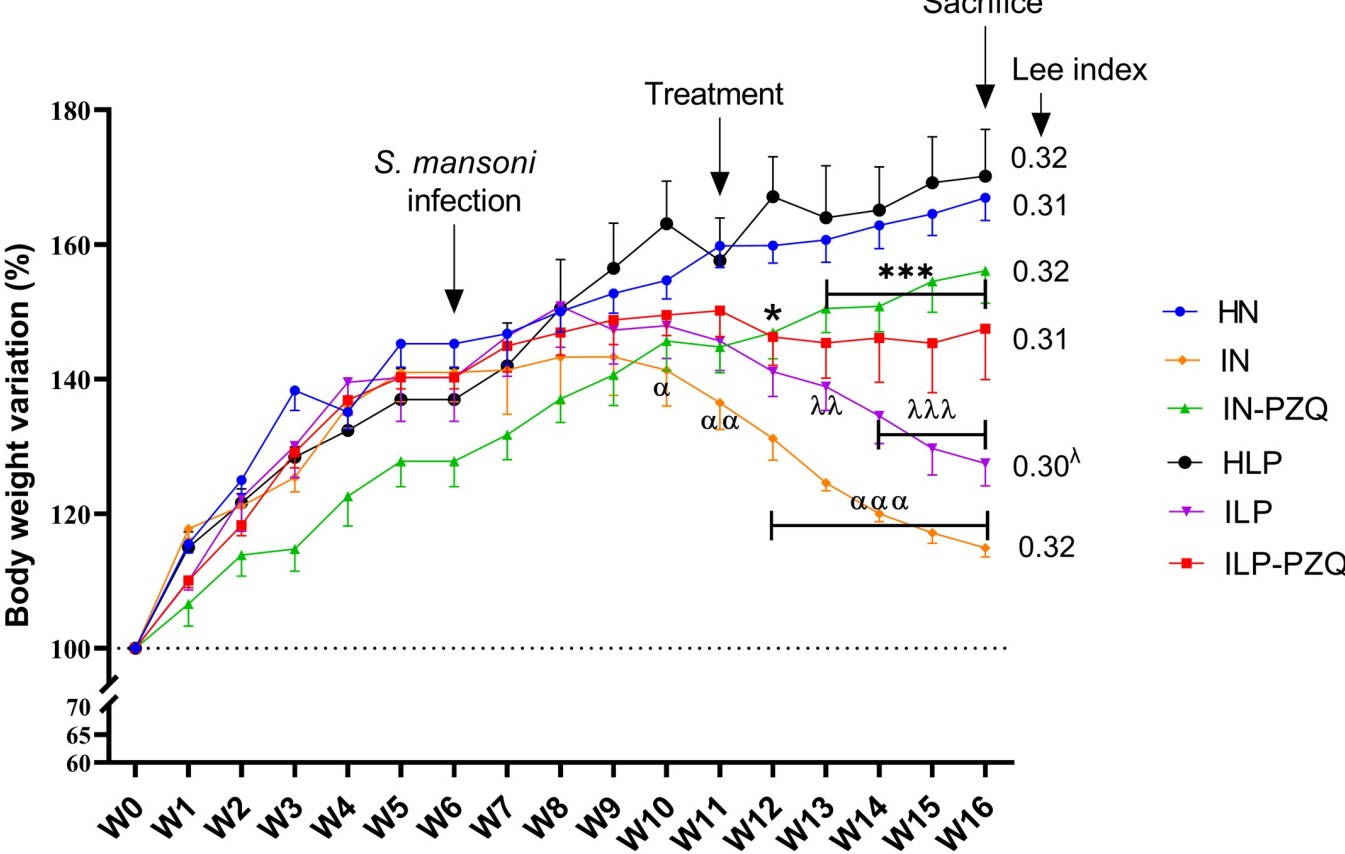

**Fig 1. Effects of praziquantel treatment on the body weight variation of *Schistosoma mansoni*-infected mice receiving a low-protein diet.** Data are expressed as mean ± SEM (n = 4–8). Group HN: healthy mice receiving a standard diet; group IN: infected-untreated mice receiving a standard diet; group IN-PZQ: infected mice receiving a standard diet and treated with praziquantel; group HLP: healthy mice receiving a low-protein diet; group ILP: infected-untreated mice receiving a low-protein diet; group ILP-PZQ: infected mice receiving a low-protein diet and treated with praziquantel. ANOVA followed by Turkey's multiple comparison test. $^{\alpha}P < 0.05$; $^{\alpha\alpha}P < 0.01$; $^{\alpha\alpha\alpha}P < 0.001$: values are significantly different from healthy mice receiving a standard diet (group HN). $^{*}P < 0.05$; $^{***}P < 0.001$: values are significantly different from the infected-untreated mice receiving a standard diet (group IN). $^{\lambda}P < 0.05$; $^{\lambda\lambda}P < 0.01$; $^{\lambda\lambda\lambda}P < 0.001$: values are significantly different from healthy mice receiving a low-protein diet (group HLP).

weight variation of mice belonging to the ILP-PZQ group wasn't statistically different from that of the ILP group at the end of the experimentation. The Lee index, an indicator of the nutritional status, did not vary after *S. mansoni* infection or after PZQ treatment of mice receiving the standard diet. However, this index was reduced in *S. mansoni* infected mice fed with a low-protein diet than in controls ($P < 0.05$). PZQ treatment did not significantly improve the body weight reduction in mice fed a low-protein diet.

 *S. mansoni* infection significantly increased mice's liver, spleen, and intestine weights receiving either the standard diet (IN) or the low-protein diet (ILP). However, the IN mice were more affected by the infection than the ILP mice, as denoted by the significant difference in hepatosplenomegaly ($P < 0.001$) and intestine enlargement ($P < 0.01$) between the two groups. Oral administration of PZQ to normally nourished and infected mice (IN-PZQ group) resulted in a significant reduction ($P < 0.001$) of the liver (32.83%), spleen (60.64%), and intestine (32.07%) weights as compared to those of infected-untreated mice. PZQ treatment then reestablished the liver, intestine and spleen weights of IN-PZQ mice as they were close to the normal range of HN naïve mice. On the contrary, PZQ treatment of infected mice receiving the low-protein diet (ILP-PZQ) did not significantly reduce the liver and intestine weight indexes that are still similar to those of ILP mice. Only the spleen weight of ILP-PZQ mice significantly decreased by 43.01% as compared to their healthy controls ($P < 0.01$) (Fig 2). Then, PZQ treatment did not improve hepatomegaly and intestine enlargement induced by the infection in mice receiving the low-protein diet.

### Praziquantel treatment is less efficacious in reducing the worm and liver egg burdens of *Schistosoma mansoni*-infected mice receiving the low-protein diet

The worm and egg burdens of mice subjected to different treatments are depicted in Fig 3. After *S. mansoni* infection, worm recovery was 47.50% for the infected-untreated mice receiving the standard diet and 45.42% for the infected-untreated mice receiving the low-protein diet. The treatment of infected mice receiving the standard diet with PZQ significantly reduced worm count by 78.95% ($P < 0.001$). In contrast, we recorded an insignificant decrease of 19.88% of worm count in the group of infected mice receiving the low-protein diet and treated with PZQ compared to their respective infected-untreated mice. PZQ treatment was ineffective in the worm burden reduction of infected mice fed the low-protein diet.

 In comparison to their respective control groups, egg burdens in the feces, the liver and the intestine were significantly reduced in either IN-PZQ mice or ILP-PZQ mice. However, the reduction of the liver egg count was more critical in infected mice receiving the standard diet (97.02%) than in the ones receiving the low-protein diet (42.50%) ($P < 0.001$).

### Praziquantel treatment failed to alleviate transaminases and gamma-glutamyl transferase activities of *Schistosoma mansoni*-infected mice receiving the low-protein diet

Administration of a low-protein diet to healthy mice (HLP) induced a significant reduction of aspartate aminotransferase (AST) activity by 47.71% as compared to healthy mice receiving a standard diet (HN). *S. mansoni* infection induced a significant increase in aspartate aminotransferase (AST), alanine aminotransferase (ALT), alkaline phosphatase (ALP) and γ-glutamyl transferase (GGT) activities, and total bilirubin concentration in mice receiving a standard diet (IN) or a low-protein diet (ILP). Oral administration of praziquantel to infected mice receiving a standard diet resulted in a significant reduction of ALT (12.53%), ALP (42.11%), and GGT (55.19%) activities, as well as total bilirubin concentration (46.12%) as

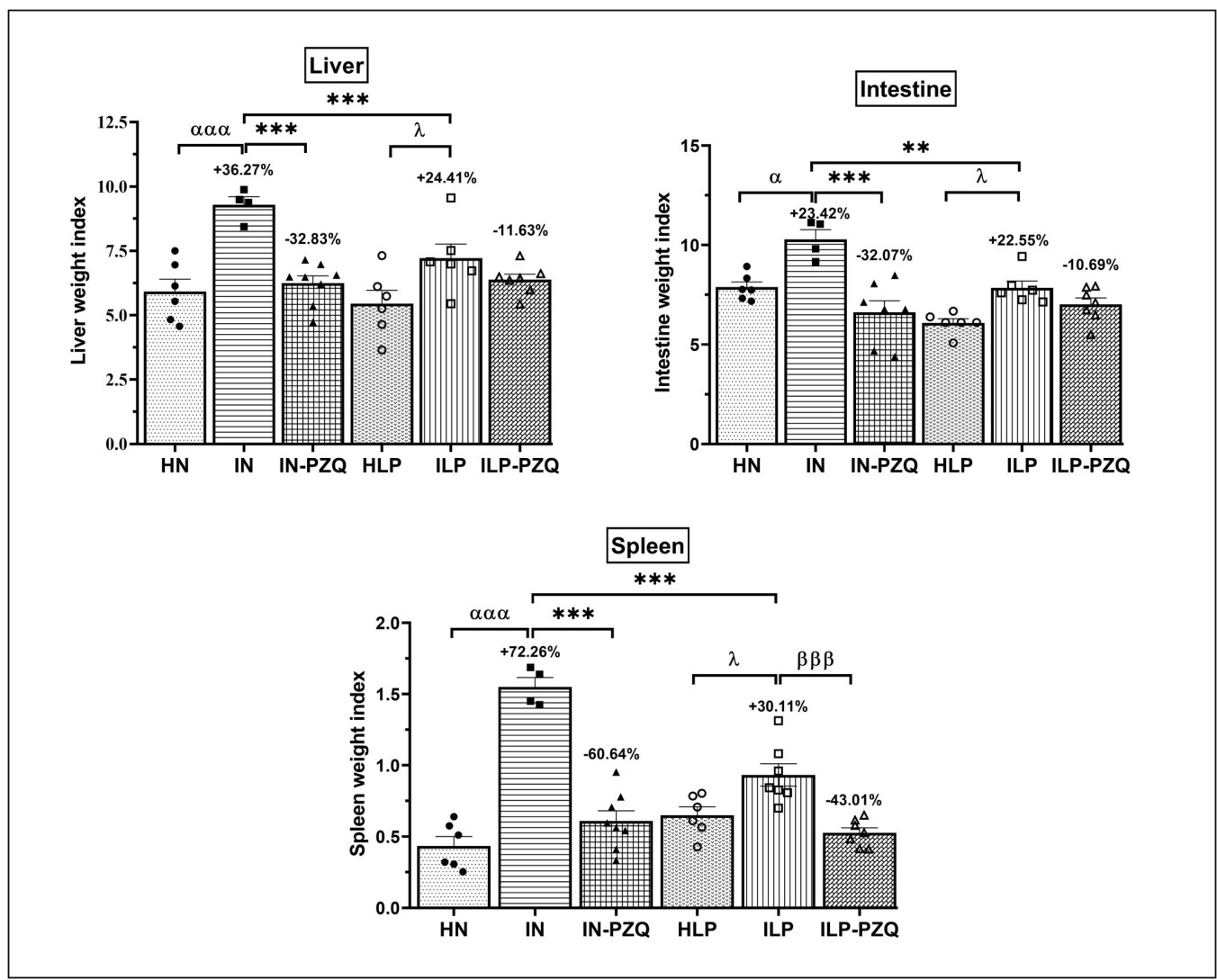

**Fig 2. Effects of praziquantel treatment on the organs weight indexes of *Schistosoma mansoni*-infected mice receiving a low-protein diet.** Data are expressed as mean ± SEM (n = 4–8). Group HN: healthy mice receiving a standard diet; group IN: infected-untreated mice receiving a standard diet; group IN-PZQ: infected mice receiving a standard diet and treated with praziquantel; group HLP: healthy mice receiving a low-protein diet; group ILP: infected-untreated mice receiving a low-protein diet; group ILP-PZQ: infected mice receiving a low-protein diet and treated with praziquantel. ANOVA followed by Turkey's multiple comparison test. $^{\alpha}P < 0.05$; $^{\alpha\alpha\alpha}P < 0.001$: values are significantly different from the healthy normal nourished animals (group HN). $^{**}P < 0.01$ $^{***}P < 0.001$: values are significantly different from the infected-untreated mice receiving a standard diet (group IN). $^{\lambda}P < 0.05$; values are significantly different from healthy mice receiving a low-protein diet (group HLP). $^{\beta\beta\beta}P < 0.001$: values are significantly different from infected-untreated mice receiving a low-protein diet (group ILP). The number on top of each bar graph represents the percentage of variation.

compared to those of their untreated controls. In infected mice receiving the low-protein diet and treated with praziquantel (ILP-PZQ), only ALP activity and total bilirubin level were reduced by 37.48% and 40.14%, respectively, when compared to those of the untreated controls. Activities of AST, ALT and GGT remained higher than those of ILP controls mice, and the differences were statistically significant for AST ($P < 0.05$) and GGT activities ($P < 0.001$) (Fig 4). PZQ treatment reduced ALP activity and total bilirubin concentration and failed to improve AST, ALT, and GGT in *S. mansoni*-infected mice receiving the low-protein diet.

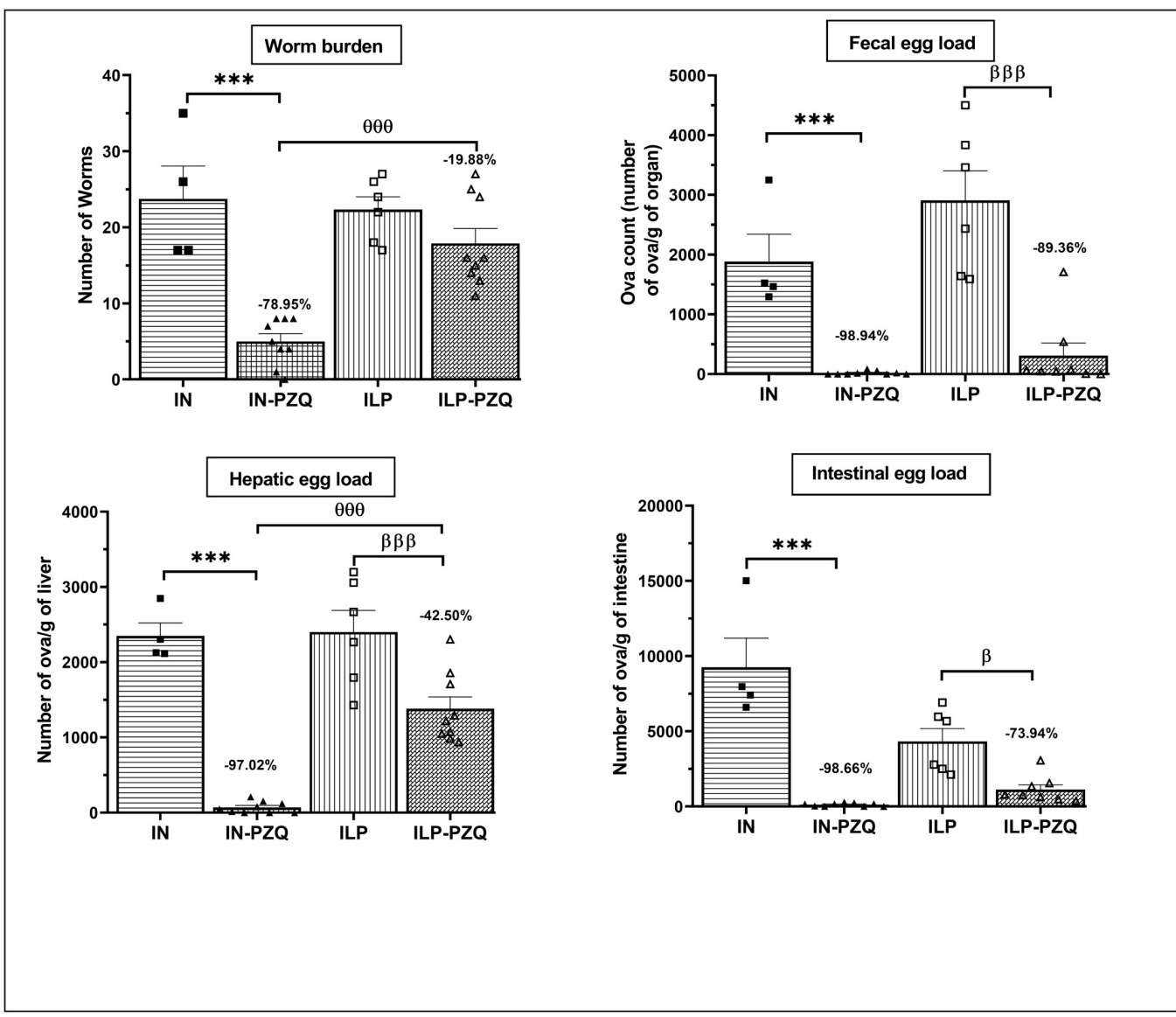

**Fig 3. Effects of praziquantel treatment on the worm and egg burdens of *Schistosoma mansoni*-infected mice receiving a low-protein diet.** Data are expressed as mean ±SEM (n = 4–9). Group HN: healthy mice receiving a standard diet; group IN: infected-untreated mice receiving a standard diet; group IN-PZQ: infected mice receiving a standard diet and treated with praziquantel; group HLP: healthy mice receiving a low-protein diet; group ILP: infected-untreated mice receiving a low-protein diet; group ILP-PZQ: infected mice receiving a low-protein diet and treated with praziquantel. ANOVA followed by Turkey multiple comparison test. ***$P < 0.001$: values are significantly different from the infected-untreated mice receiving a standard diet (group IN). ${}^{\beta}P < 0.05$; ${}^{\beta\beta\beta}P < 0.001$: values are significantly different from infected-untreated mice receiving a low-protein diet (group ILP). ${}^{\theta\theta\theta}P < 0.001$: values significantly different from infected mice receiving a standard diet and treated with praziquantel at 100 mg/kg for 5 consecutive days (group IN-PZQ). The number on top of each bar graph represents the percentage of variation.

## Praziquantel treatment does not enhance the biomarkers of the general metabolism of *Schistosoma mansoni*-infected mice receiving the low-protein diet

As shown in Fig 5, the low-protein diet significantly decreased the total proteins, albumin, glucose, and calcium concentrations in HLP mice compared to HN mice. When infected, mice submitted to the standard or the low-protein diet (IN or ILP group) showed significant

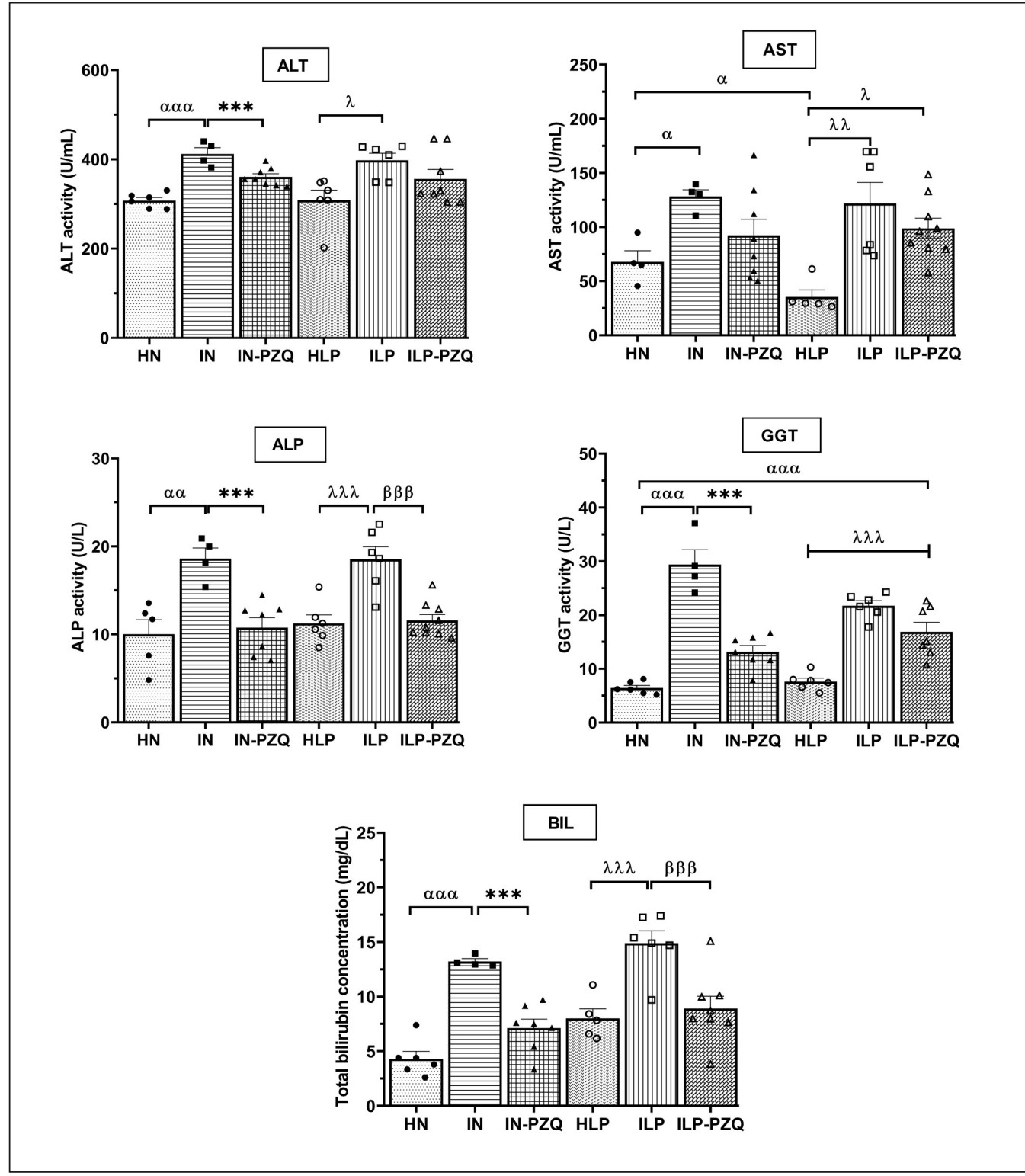

**Fig 4. Effects of praziquantel treatment on the liver function biomarkers of *Schistosoma mansoni*-infected mice receiving a low-protein diet.** Data are expressed as mean ±SEM (n = 4–8). AST: aspartate aminotransferase; ALAT: alanine aminotransferase; ALP: alkaline phosphatase; GGT: gamma-glutamyl transferase; BIL: total bilirubin. Group HN: healthy mice receiving a standard diet; group IN: infected-untreated mice receiving a standard diet; group IN-PZQ: infected mice receiving a standard diet and treated with praziquantel; group HLP: healthy mice receiving a low-protein diet; group ILP: infected-untreated mice receiving a low-protein diet; group ILP-PZQ: infected mice receiving a low-protein diet and treated with praziquantel. ANOVA followed by

Turkey multiple comparison test. $^{a}P < 0.05$; $^{aa}P < 0.01$; $^{aaa}P < 0.001$: values are significantly different from healthy mice receiving a standard diet (group HN). $^{***}P < 0.001$: values are significantly different from infected-untreated mice receiving a standard diet (group IN). $^{\lambda\lambda\lambda}P < 0.001$: values are significantly different from healthy mice receiving a low-protein diet (group HLP). $^{\beta\beta\beta}P < 0.001$: values are significantly different from infected-untreated mice receiving a low-protein diet (group ILP).

decreases in total proteins, albumin, glucose, and calcium levels and a significant increase in iron concentration compared to their respective controls (HN and HLP groups). Administration of PZQ to *S. mansoni*-infected mice receiving the standard diet did not reestablish the expected concentration of proteins, albumin, and glucose. Still, it significantly restored the iron and calcium concentrations ($P < 0.001$). Compared to ILP mice, we recorded significant increases of total proteins level by 35.54% and glucose concentration by 52.94% in ILP-PZQ mice. However, these increases were not enough to get these concentrations close to normal ones since the total proteins, albumin, and glucose levels were still significantly lower in ILP-PZQ mice than in HN mice ($P < 0.05$ and $P < 0.001$). Albumin, iron and calcium concentrations of ILP-PZQ mice did not improve after PZQ treatment. The iron level of ILP-PZQ mice significantly remained higher than that of IN-PZQ mice ($P < 0.001$). At the same time, their calcium concentration was low compared to that of the IN-PZQ mice ($P < 0.05$). PZQ treatment ameliorated total proteins and glucose levels in infected mice receiving the low-protein diet but failed to normalize them compared to healthy mice receiving the standard diet. Moreover, PZQ treatment did not improve albumin, iron and calcium levels in these mice.

## PZQ treatment restores the biomarkers levels of lipid metabolism of *Schistosoma mansoni*-infected mice receiving the low-protein diet

Compared to HN mice, administration of a low-protein diet to mice (HLP) induced a significant decrease in serum HDL cholesterol level by 18.24% and triglycerides by 28.74%. *S. mansoni*-infected mice receiving the standard or the low-protein diet showed a significantly lower concentration of total cholesterol, LDL cholesterol, and triglycerides than their uninfected controls. When treating mice with PZQ, total cholesterol, LDL cholesterol, and triglycerides concentrations were significantly improved by 14.53%, 50.35%, and 38.79%, respectively, for the IN-PZQ group and by 24.64%, 59.16%, and 42.53% for the ILP-PZQ group. In addition, a significant increase in HDL cholesterol ($P < 0.01$) was noted in ILP-PZQ mice as compared to ILP ones (Fig 6). PZQ treatment improved total cholesterol, LDL cholesterol and triglycerides in *S. mansoni*-infected mice receiving the standard or the low-protein diet.

## Praziquantel treatment does not restore the concentration of red blood cells of *Schistosoma mansoni*-infected mice receiving a low-protein diet

Administration of a low-protein diet to healthy mice (HLP) resulted in a significant reduction of 45.29% of the red blood cell count ($P < 0.001$) as compared to healthy mice receiving a standard diet (HN). *S. mansoni* infection significantly reduced the red blood cell count, hematocrit, and lymphocytes. At the same time, we recorded an increase in white blood cell count and eosinophil percentage in IN mice compared to HN mice. In infected-untreated mice receiving the low-protein diet (ILP), we noted a decrease in lymphocyte percentage ($P < 0.001$) but an increase of white blood cell counts ($P < 0.001$) as compared to healthy mice of the HLP group. Only one mouse expressed eosinophilia in this group. Treatment with praziquantel restored the hematocrit, the total leukocytes, lymphocytes, and eosinophils percentage in mice receiving the standard or the low-protein diet compared to their controls. The red blood cell concentration of all the infected mice receiving praziquantel remained significantly lower than that of IN mice ($P < 0.01$ for IN-PZQ and $P < 0.001$ for ILP-PZQ) (Table 4).

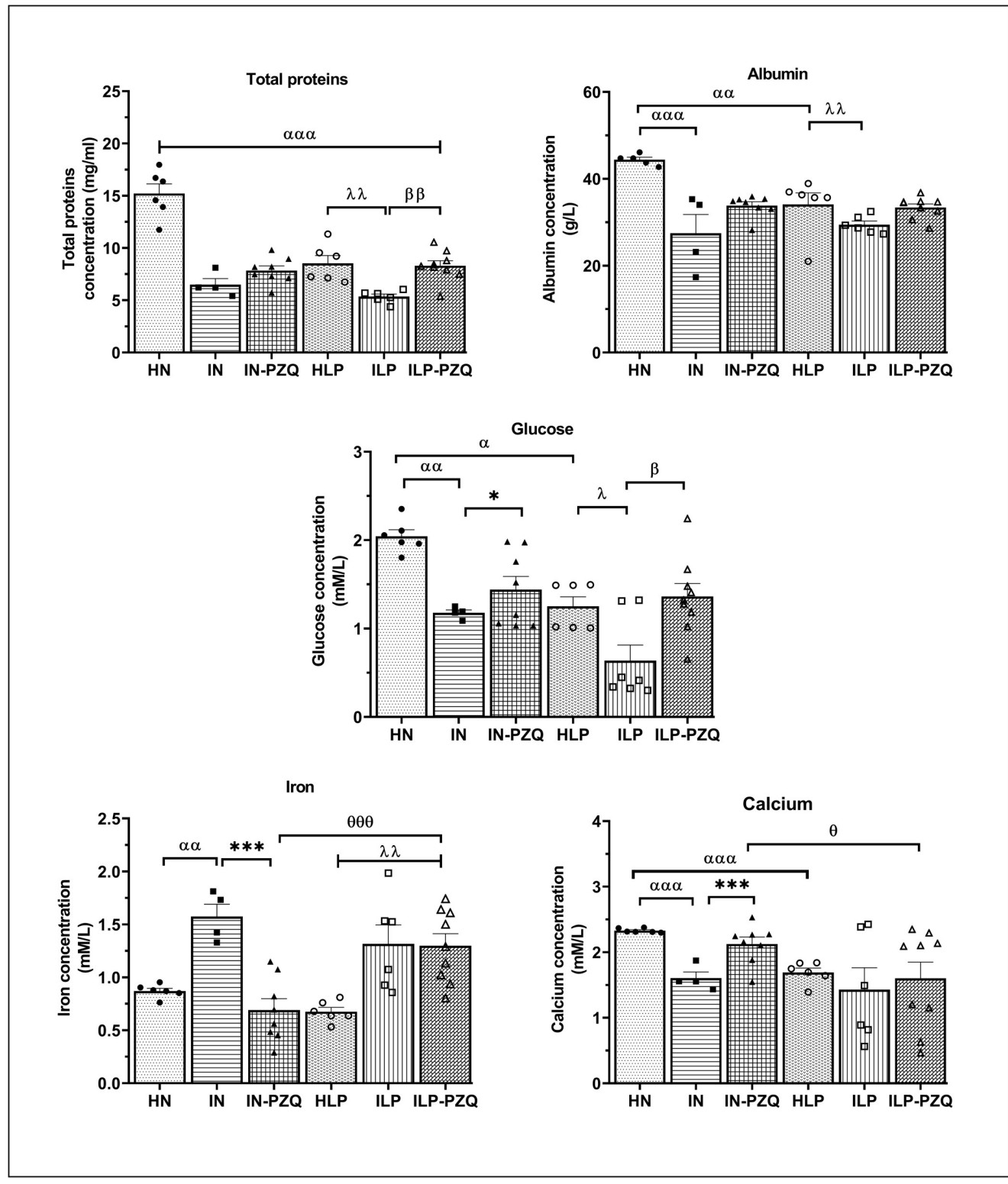

**Fig 5. Effects of praziquantel treatment on some biomarkers of the general metabolism in *Schistosoma mansoni*-infected mice receiving a low-protein diet.** Data are expressed as mean ±SEM (n = 4–8). Group HN: healthy mice receiving a standard diet; group IN: infected-untreated mice receiving a standard diet; group IN-PZQ: infected mice receiving a standard diet and treated with praziquantel; group HLP: healthy mice receiving a low-protein diet; group ILP: infected-untreated mice receiving a low-protein diet; group ILP-PZQ: infected mice receiving a low-protein diet and treated with praziquantel. ANOVA followed by Turkey multiple comparison test. $^{\alpha}P < 0.05$; $^{\alpha\alpha}P < 0.01$; $^{\alpha\alpha\alpha}P < 0.001$: values are significantly different from healthy mice receiving a standard diet (group HN). $^{***}P < 0.001$: values are significantly different from infected-untreated mice receiving a standard diet (group IN). $^{\lambda}P < 0.05$; $^{\lambda\lambda}P < 0.01$ values are significantly different from healthy mice receiving a low-protein diet (group HLP). $^{\beta}P < 0.05$ $^{\beta\beta}P < 0.01$: values are significantly

different from infected-untreated mice receiving a low-protein diet (group ILP). $^{\theta}P < 0.05$; $^{\theta\theta\theta}P < 0.001$: values significantly different from infected mice receiving a standard diet and treated with praziquantel at the dose of 100 mg/kg for 5 consecutive days (group IN-PZQ).

### Praziquantel treatment is less efficacious in normalizing malondialdehyde concentration of *Schistosoma mansoni*-infected mice receiving the low-protein diet

As shown in Fig 7, administration of a low-protein diet to mice (ILP group) did not significantly modify oxidative stress biomarkers compared to healthy mice receiving a standard diet

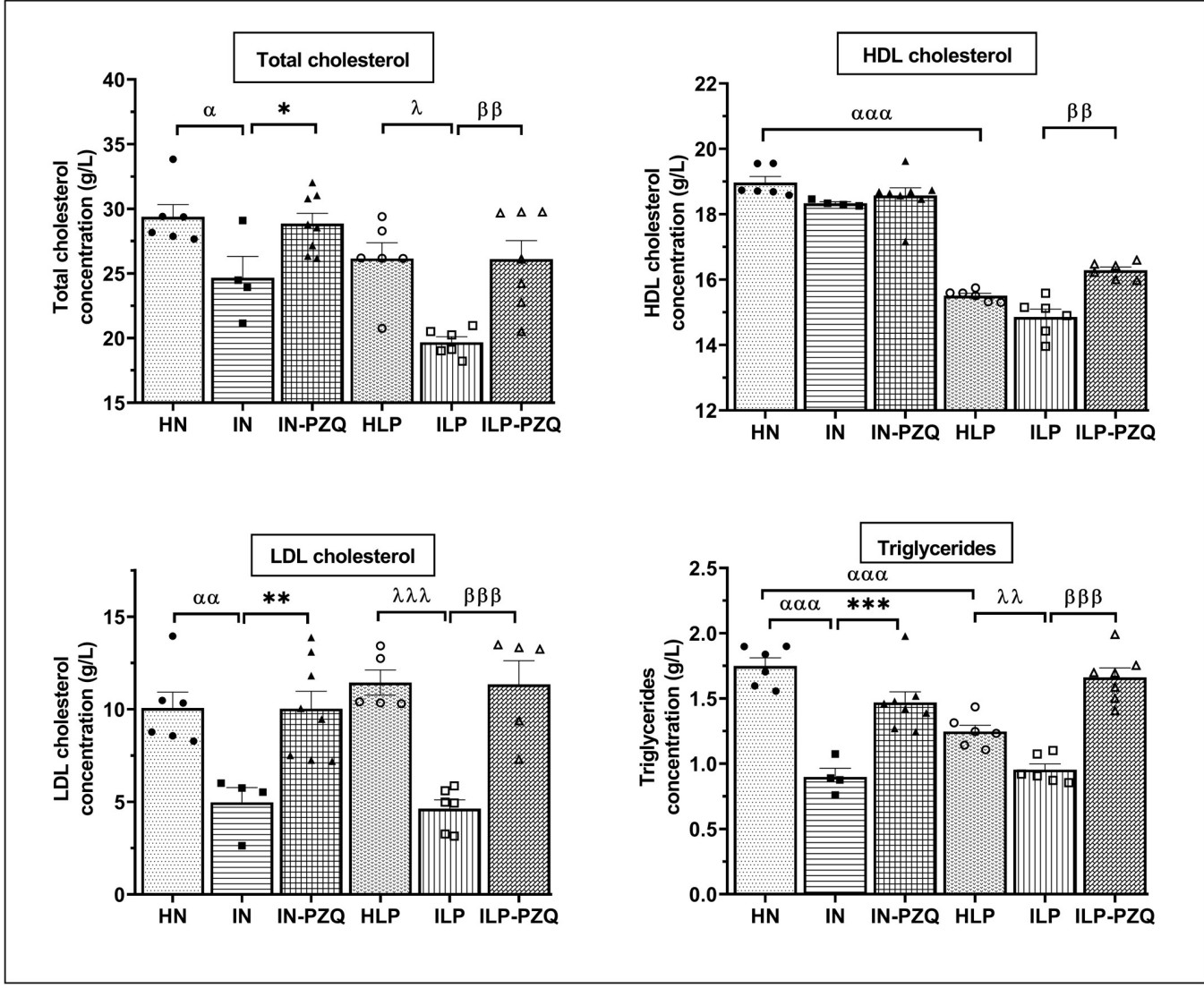

**Fig 6. Effects of praziquantel treatment on the lipid profile of *Schistosoma mansoni*-infected mice receiving a low-protein diet.** Data are expressed as mean ±SEM (n = 4–8). Group HN: healthy mice receiving a standard diet; group IN: infected-untreated mice receiving a standard diet; group IN-PZQ: infected mice receiving a standard diet and treated with praziquantel; group HLP: healthy mice receiving a low-protein diet; group ILP: infected-untreated mice receiving a low-protein diet; group ILP-PZQ: infected mice receiving a low-protein diet and treated with praziquantel. ANOVA followed by Turkey multiple comparison test. $^{\alpha}P < 0.05$; $^{\alpha\alpha}P < 0.01$; $^{\alpha\alpha\alpha}P < 0.001$: values are significantly different from healthy mice receiving a standard diet (group HN). $^{*}P < 0.05$; $^{**}P < 0.01$; $^{***}P < 0.001$: values are significantly different from infected-untreated mice receiving a standard diet (group IN). $^{\lambda}P < 0.05$; $^{\lambda\lambda}P < 0.01$ $^{\lambda\lambda\lambda}P < 0.001$: values are significantly different from healthy mice receiving a low-protein diet (group HLP). $^{\beta\beta}P < 0.01$; $^{\beta\beta\beta}P < 0.001$: values are significantly different from infected-untreated mice receiving a low-protein diet (group ILP).

**Table 4. Effects of praziquantel treatment on blood parameters in *Schistosoma mansoni*-infected mice receiving a low-protein diet.**

| Parameters | HN | IN | IN-PZQ | HLP | ILP | ILP-PZQ |
|---|---|---|---|---|---|---|
| Red blood cells ($10^6$/µL) | 8.43 ± 0.56 | 3.92 ± 0.45 [αα] | 5.54 ± 0.84[αα] | 4.61 ± 1.21[ααα] | 3.44 ± 0.27[ααα] | 4.67 ± 0.20[ααα] |
| Hemoglobin (g/dL) | 13.19 ± 0.33 | 11.50 ± 0.79 | 13.88 ± 0.71 | 12.66 ± 0.94 | 10.34 ± 0.58 | 12.86 ± 0.57 |
| Hematocrit (%) | 42.92 ± 2.39 | 33.90 ± 3.49[**α**] | 48.46 ± 2.35[**] | 42.36 ± 4.12 | 34.77 ± 3.27 | 42.20 ± 2.37 |
| White blood cells ($10^3$/µL) | 6.64 ± 0.86 | 9.95 ± 1.01[**α**] | 5.74 ± 0.63[**] | 4.15 ± 0.42 | 9.66 ± 0.78[λ] | 4.24 ± 0.37[β] |
| Lymphocytes (%) | 84.90 ± 0.88 | 42.48 ± 3.50[αααα] | 80.60 ± 4.52[**] | 65.18 ± 9.79 | 19.74 ± 2.00[λλλ] | 66.07 ± 7.25[ββ] |
| Eosinophils (%) | 0.10 ± 0.06 | 1.90 ± 0.30[αα] | 0.05 ± 0.05[**] (1) | 0.17 ± 0.17 (1) | 0.05 ± 0.05 (1) | 0.68 ± 0.45 |
| Platelets ($10^3$/µL) | 522.80 ± 78.35 | 224.20 ±16.77 | 309.00 ± 15.70 | 514.00 ± 87.46 | 158.40 ± 18.08 | 250.40 ± 32.15 |

Data are expressed as mean ±SEM (n = 4–8). MCH: mean corpuscular hemoglobin. Group HN: healthy mice receiving a standard diet; group IN: infected-untreated mice receiving a standard diet; group IN-PZQ: infected mice receiving a standard diet and treated with praziquantel; group HLP: healthy mice receiving a low-protein diet; group ILP: infected-untreated mice receiving a low-protein diet; group ILP-PZQ: infected mice receiving a low-protein diet and treated with praziquantel. ANOVA followed by Turkey multiple comparison test.

[α] $P < 0.05$

[αα] $P < 0.01$

[ααα] $P < 0.001$: values are significantly different from healthy mice receiving a standard diet (group HN).

[**] $P < 0.01$: values are significantly different from infected-untreated mice receiving a standard diet (group IN).

[λ] $P < 0.05$

[λλλ] $P < 0.001$: values are significantly different from healthy mice receiving a low-protein diet (group HLP).

[β] $P < 0.05$

[ββ] $P < 0.01$: values are significantly different from infected-untreated mice receiving a low-protein diet (group ILP).

(HN). In the IN and ILP groups of mice, malondialdehyde (MDA) concentration significantly increased. In contrast, superoxide dismutase (SOD) and catalase (CAT) activities and reduced glutathione (GSH) concentration significantly decreased as compared to their respective controls. The nitrites level decreased only in IN group compared to the HN group. Praziquantel treatment ameliorated the MDA concentration by reducing it by 64.73% in IN-PZQ as compared to IN ($P < 0.001$) and 25.72% in ILP-PZQ as compared to ILP ($P < 0.05$). Despite this significant reduction, the MDA concentration of ILP-PZQ mice was 69.56% higher than that of IN-PZQ mice ($P < 0.05$).

In both standard and low-protein diet-feed infected mice, treatment with praziquantel restores the SOD and CAT activities. We recorded 49.35% and 63.96% increase of SOD activity and 33.29% and 57.22% increases of catalase activity in IN-PZQ and ILP-PZQ groups of mice, respectively, when compared to their infected-untreated controls groups. However, praziquantel treatment did not reverse the reduced concentration of GSH in IN-PZQ mice. On the contrary, GSH concentration significantly increased by 47.34% in ILP-PZQ mice compared to ILP mice ($P < 0.001$). Furthermore, PZQ treatment did not restore the nitrite levels, whatever the type of diet, and it was significantly low in ILP-PZQ mice compared to ILP mice ($P < 0.05$), as well as to IN-PZQ mice ($P < 0.05$). PZQ treatment improved the MDA concentration and SOD and CAT activities of IN-PZQ and ILP-PZQ mice but did not bring the MDA concentration of ILP-PZQ mice near the normal range.

## PZQ treatment restores the cytokines Th1, Th2, Th17, and Treg levels, except IL-5, of *Schistosoma mansoni*-infected mice receiving the low-protein diet

Cytokines Th1, Th2, Th17, and Treg levels were measured in the sera of mice after praziquantel treatment and are illustrated in Fig 8. The diet did not influence the immune response since

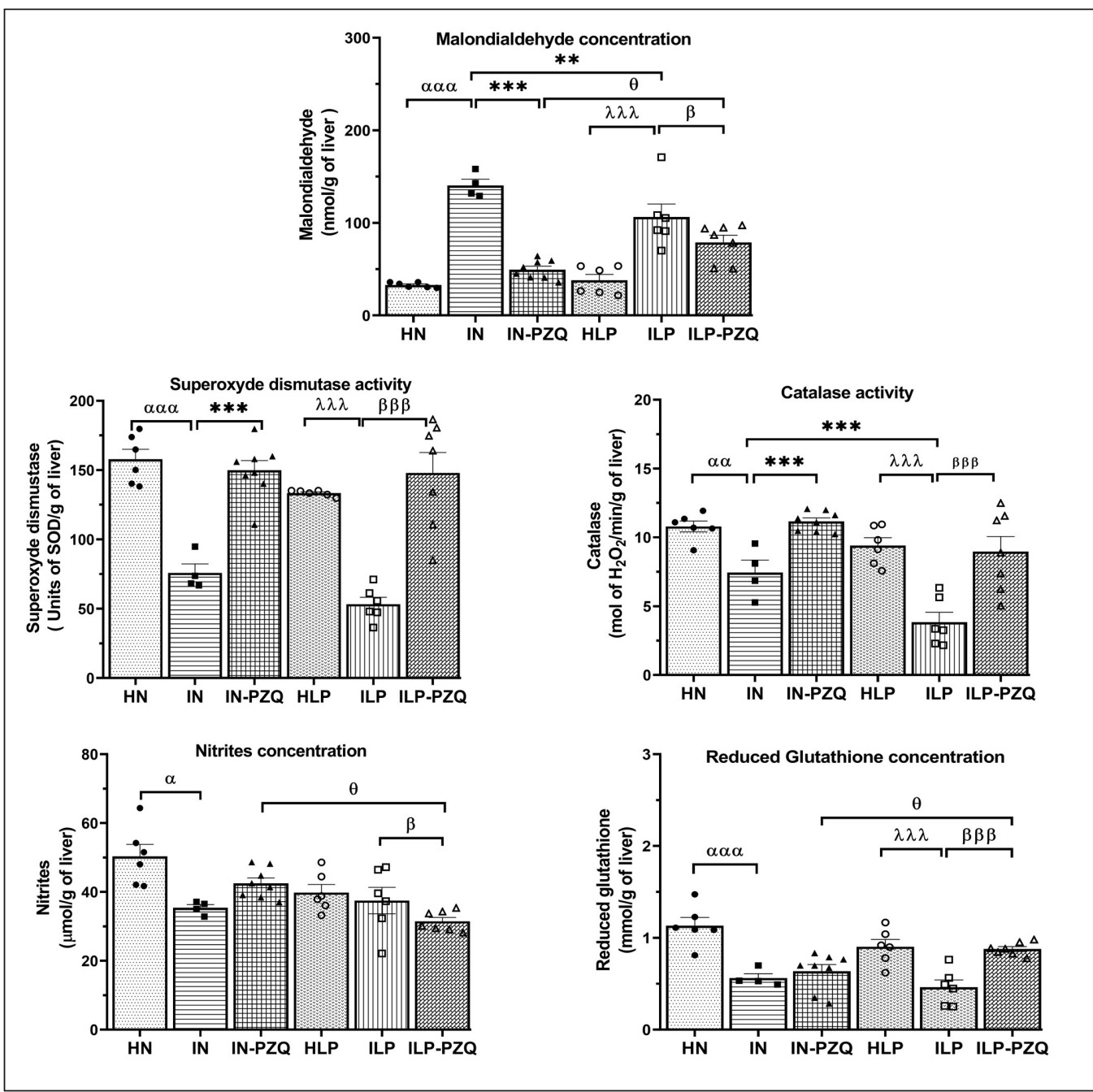

**Fig 7. Effects of praziquantel treatment on the liver oxidative stress biomarkers of *Schistosoma mansoni*-infected mice receiving a low-protein diet.** Data are expressed as mean ±SEM (n = 4–8). Group HN: healthy mice receiving a standard diet; group IN: infected-untreated mice receiving a standard diet; group IN-PZQ: infected mice receiving a standard diet and treated with praziquantel; group HLP: healthy mice receiving a low-protein diet; group ILP: infected-untreated mice receiving a low-protein diet; group ILP-PZQ: infected mice receiving a low-protein diet and treated with praziquantel. ANOVA followed by Turkey multiple comparison test. $^{\alpha}P < 0.05$; $^{\alpha\alpha}P < 0.01$; $^{\alpha\alpha\alpha}P < 0.001$: values are significantly different from healthy mice receiving a standard diet (group HN). $^{**}P < 0.01$; $^{***}P < 0.001$: values are significantly different from infected-untreated mice receiving a standard diet (group IN). $^{\lambda\lambda\lambda}P < 0.001$: values are significantly different from healthy mice received a low-protein diet (group HLP). $^{\beta}P < 0.05$; $^{\beta\beta\beta}P < 0.001$: values are significantly different from infected-untreated mice receiving a low-protein diet (group ILP). $^{\theta}P < 0.05$: values significantly different from infected mice receiving a standard diet and treated with praziquantel at the dose of 100 mg/kg for 5 consecutive days (group IN-PZQ).

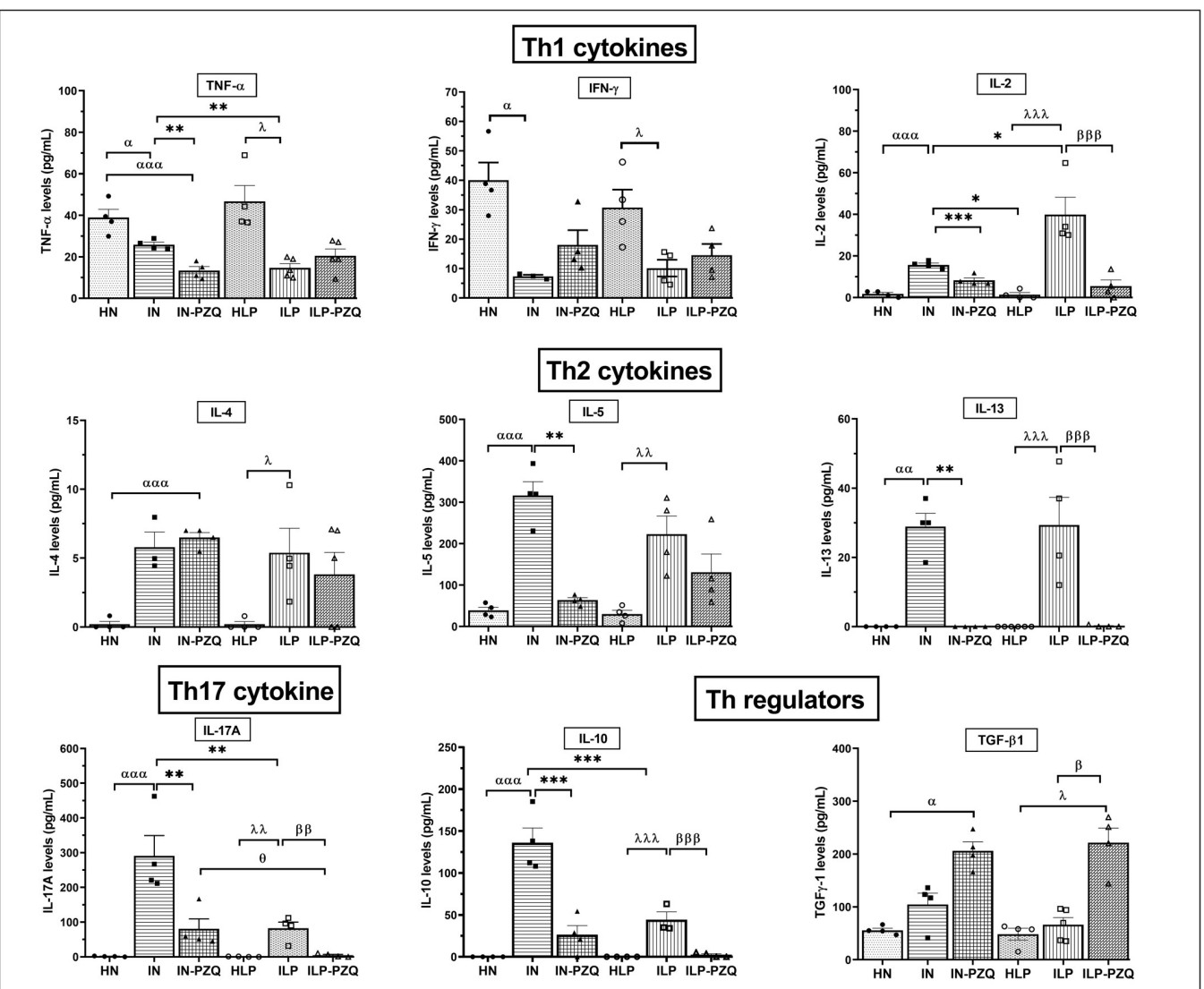

**Fig 8. Effects of praziquantel treatment on cytokines production of *Schistosoma mansoni*-infected mice receiving a low-protein diet.** Data are expressed as mean ±SEM (n = 4–8). Group HN: healthy mice receiving a standard diet; group IN: infected-untreated mice receiving a standard diet; group IN-PZQ: infected mice receiving a standard diet and treated with praziquantel; group HLP: healthy mice receiving a low-protein diet; group ILP: infected-untreated mice receiving a low-protein diet; group ILP-PZQ: infected mice receiving a low-protein diet and treated with praziquantel. ANOVA followed by Turkey multiple comparison test. $^{\alpha}P < 0.05$; $^{\alpha\alpha}P < 0.01$; $^{\alpha\alpha\alpha}P < 0.001$: values are significantly different from healthy mice receiving a standard diet (group HN). $^{*}P < 0.05$; $^{**}P < 0.01$; $^{***}P < 0.001$: values are significantly different from infected-untreated mice receiving a standard diet (group IN). $^{\lambda}P < 0.05$; $^{\lambda\lambda}P < 0.01$; $^{\lambda\lambda\lambda}P < 0.001$: values are significantly different from healthy mice received a low-protein diet (group HLP). $^{\beta}P < 0.05$; $^{\beta\beta}P < 0.01$; $^{\beta\beta\beta}P < 0.001$: values are significantly different from infected-untreated mice receiving a low-protein diet (group ILP). $^{\theta}P < 0.05$: values significantly different from infected mice receiving a standard diet and treated with praziquantel at the dose of 100 mg/kg for 5 consecutive days (group IN-PZQ).

no variation in cytokine levels was recorded between healthy mice receiving the standard diet and healthy mice receiving the protein-deficient diet.

In *S. mansoni*-infected mice, serum levels of Th1 cytokines varied significantly. A 32.27% and 68.53% reduction for TNF-α and a 81.61% and 66.96% reduction for IFN-γ were recorded in IN and ILP mice, respectively, compared to their respective healthy controls. The TNF-α reduction was more critical in the ILP group than in IN group ($P < 0.01$). The level of IL-2 rose substantially higher in IN and ILP mice, 8.23-fold and 38.39-fold, respectively, compared to their healthy controls (Fig 8). Praziquantel treatment did not restore TNF-α and IFN-γ

levels, whatever the type of the diet for mice. On the contrary, this treatment significantly reversed the increased IL-2 levels of infected mice, as we recorded a 47% and 89.57% reduction in mice of groups IN-PZQ and ILP-PZQ, respectively.

As denoted in Fig 8, IL-4 was poorly detected in the sera of healthy mice, and IL-13 was absent. Whatever the type of diet, serum concentrations of IL-4, IL-5, and IL-13 significantly increased after *S. mansoni* infection. Praziquantel treatment completely abrogated infection-induced IL-13 levels in both treatment groups (IN-PZQ and ILP-PZQ). When treated with praziquantel, infected mice receiving the standard diet showed an 81.31% reduction of IL-5 level ($P < 0.01$) compared to their infected-untreated controls. In contrast, treatment did not significantly affect infected mice receiving a low-protein diet. In addition, the praziquantel treatment did not significantly reduce the high IL-4 level of infected mice in the IN-PZQ and ILP-PZQ groups.

IL-17A and IL-10 were not detected in the sera of healthy mice receiving a standard or a low-protein diet. However, *S. mansoni* infected mice secreted significant levels of IL-17A and IL-10, particularly in infected mice receiving the standard diet. IL-17A and IL-10 were 75.40% and 69.95%, respectively, lower in the ILP group than in the IN group. The administration of praziquantel to infected mice significantly reduced IL-17A levels by 71.45% and 94.08% and IL-10 levels by 82.27% and 94.71% for IN-PZQ and ILP-PZQ groups, respectively, as compared to their infected untreated controls. The level of TGF-β1 was not modified by *S. mansoni* infection, whatever the type of diet administered to mice. However, it increased significantly after praziquantel treatment, to 1.81-fold and 3.05-fold in IN-PZQ and ILP-PZQ groups, respectively, compared to their healthy controls (Fig 8). PZQ treatment restores the cytokines Th1, Th2, Th17, and Treg levels of infected mice receiving the standard or the low-protein diet, except for the IL-5 level of infected mice receiving the low-protein diet.

## Praziquantel treatment does not restore the CXCL-10/IP-10 and CCL3/MIP-1α gene expression of *Schistosoma mansoni*-infected mice receiving the low-protein diet

The diet did not induce any variation of the mRNA expression of cytokines and chemokines in healthy mice (HN), except for the gene expression of FGF-1 and CCL3, which are low in HLP mice. *S. mansoni* infection significantly decreased the mRNA expression of TGF-β1 by 81.71% and 79.68%, and FGF-1 by 79.25% and 64.45% in the standard diet feed (IN) and low-protein diet-feed mice (ILP), respectively, as compared to their healthy controls. We also recorded significant reductions of mRNA expression of chemokines CCL2/MCP-1, CCL3/MIP-1α, and CXCL-10/IP-10. In addition, we noted a decrease of IFN-γ mRNA expression and an increase of FoxP3 IL-10 and IL-13 mRNA expression in the liver of infected mice compared to healthy mice, although it was not statistically significant. Oral administration of praziquantel to infected mice completely reversed the diminished TGF-β1 gene expression. It was 3.70-fold and 2.85-fold higher in infected mice receiving the standard diet (IN-PZQ) or the low-protein diet (ILP-PZQ), respectively, than in their infected untreated controls. Praziquantel treatment did not restore the mRNA expression of FGF-1 and MCP-1 in infected mice, whatever their diet. Regarding the gene expression of chemokines, significant increases of the mRNA expression of CXCL-10/IP-10 and CCL3/MIP-1α were 40.59-fold and 5.96-fold, respectively, higher in IN-PZQ mice than those of IN mice. However, their levels did not change in infected mice fed with a low-protein diet and treated with praziquantel (ILP-PZQ) (Fig 9). Finally, PZQ treatment restored the gene expression of TGF-β1 but failed to do it for CXCL-10/IP-10 and CCL3/MIP-1α in infected mice fed with a low-protein diet.

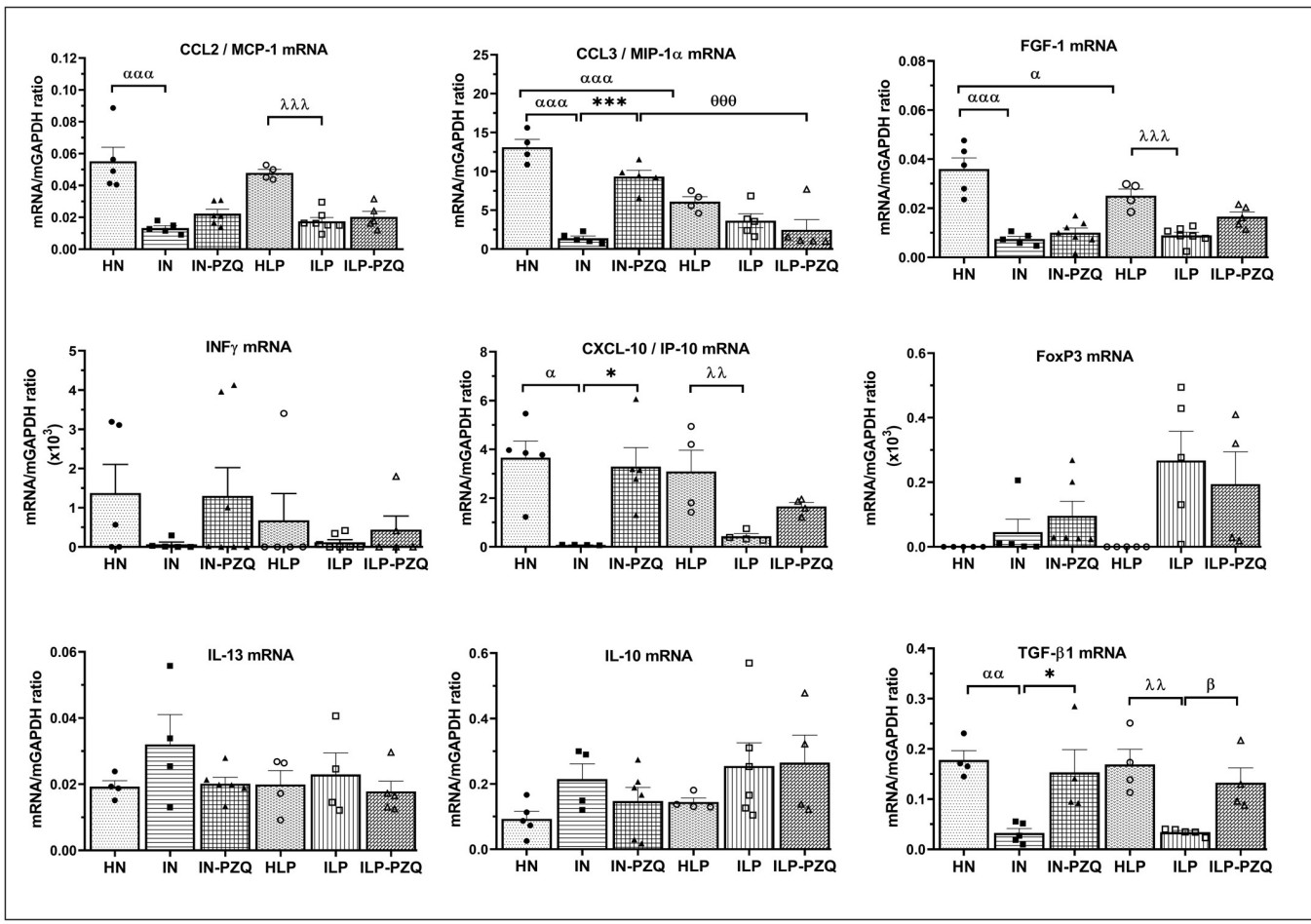

**Fig 9. Effects of praziquantel treatment on the mRNA expression of cytokines and chemokines of *Schistosoma mansoni*-infected mice receiving a low-protein diet.** Data are expressed as mean ±SEM (n = 4–8). Group HN: healthy mice receiving a standard diet; group IN: infected-untreated mice receiving a standard diet; group IN-PZQ: infected mice receiving a standard diet and treated with praziquantel; group HLP: healthy mice receiving a low-protein diet; group ILP: infected-untreated mice receiving a low-protein diet; group ILP-PZQ: infected mice receiving a low-protein diet and treated with praziquantel. ANOVA followed by Turkey multiple comparison test. $^{\alpha}P < 0.05$; $^{\alpha\alpha}P < 0.01$; $^{\alpha\alpha\alpha}P < 0.001$: values are significantly different from healthy mice receiving a standard diet (group HN). $^{*}P < 0.05$; $^{**}P < 0.001$: values are significantly different from infected-untreated mice receiving a standard diet (group IN). $^{\lambda\lambda}P < 0.01$; $^{\lambda\lambda\lambda}P < 0.001$: values are significantly different from healthy mice received a low-protein diet (group HLP). $^{\beta}P < 0.05$; values are significantly different from infected-untreated mice receiving a low-protein diet (group ILP). $^{\theta\theta\theta}P < 0.001$: values significantly different from infected mice receiving a standard diet and treated with praziquantel at the dose of 100 mg/kg for 5 consecutive days (group IN-PZQ).

## Praziquantel treatment does not reverse the liver and intestine histological injuries in *Schistosoma mansoni*-infected mice receiving a low-protein diet

The qualitative analysis of the liver and the intestine sections of HLP mice did not substantially differ from HN mice. Histological examination of hematoxylin-eosin (H&E) and picrosirius (PS) stained liver sections showed a typical portal triad and hepatic lobules with normal hepatocytes radiating from the central vein to the periphery of the lobule (Fig 10A, 10A', 10D and 10D'). Intestine sections showed normal muscularis mucosa, lamina propria, and epithelial layer. The villous height or crypt depth was typical (Fig 11A, 11A', 11D and 11D').

The liver sections of *S. mansoni*-infected mice receiving the standard or the low-protein diet revealed a granulomatous inflammatory response around the parasite egg, with a considerable density of collagen fibers. Signs of reactional hepatitis in mice, represented by infiltration of mononuclear leukocytes mixed with polymorphonuclear eosinophils within the portal

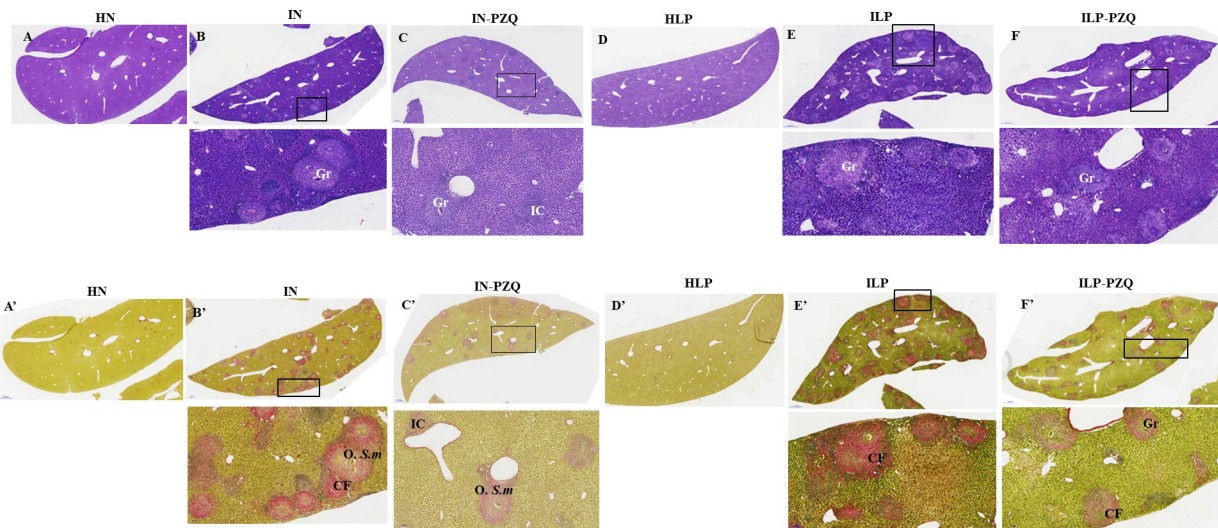

**Fig 10. Effects of praziquantel treatment on the liver histology of *Schistosoma mansoni*-infected mice receiving a low-protein diet.** A, B, C, D E, and F: Hematoxylin-eosin staining A', B', C', D', E' and F': Picrosirius staining. Group IN: infected-untreated mice receiving a standard diet; group IN-PZQ: infected mice receiving a standard diet and treated with praziquantel; group ILP: infected-untreated mice receiving a low-protein diet; group ILP-PZQ: infected mice receiving a low-protein diet and treated with praziquantel. IC: inflammatory cells, Gr: granuloma, O. S.m: egg of *S. mansoni*, CF: collagen fiber.

spaces and around the central vein, were observed (Fig 10B, 10B', 10E and 10E'). The liver sections of mice treated with praziquantel (IN-PZQ and ILP-PZQ groups) showed a low density of granulomas with slight collagen deposition. The liver sections were nearly free of granulomas for some animals in both groups. However, some sinusoids were still dilated and infiltrated with lymphocytes (Fig 10C, 10C', 10F and 10F'). These damages were primarily observed in infected mice receiving a low-protein diet and treated with praziquantel (ILP-PQZ).

Histopathological examination of the intestine sections of both IN and ILP mice revealed chronic inflammation and giant granulomas in the mucosa, the submucosa, and sometimes in the muscular layer. We also observed muscularis thickening, villous atrophy, crypt hyperplasia, and cell hypertrophy (Fig 11B, 11B', 11E and 11E'). The intestine structure of IN-PZQ and ILP-PZQ mice) was ameliorated as granulomas were few and immature. In most cases, there was neither egg with concentric collagen fibers nor an accumulation of epithelioid cells and lymphocytes in the intestine sections. However, in the ILP-PZQ group, some sinusoids were still dilated and infiltrated with lymphocytes on the liver sections.

## Praziquantel treatment is inefficacious in reducing the volume of hepatic granulomas in *Schistosoma mansoni*-infected mice receiving the low-protein diet

The number of granulomas per field was $0.45 \pm 0.01$ and $0.51 \pm 0.05$ in the liver and $0.60 \pm 0.08$ and $0.50 \pm 0.10$ in the intestine of infected mice receiving the standard diet (IN) or the low-protein diet (ILP), respectively. Praziquantel treatment significantly reduced the granulomas number in the liver of infected mice receiving the standard or the protein-deficient diet by 44.49% and 33.16%, respectively, compared to their infected-untreated controls (Fig 12A). Moreover, praziquantel treatment significantly decreases by 49.43% the number of intestine granulomas per field in the IN-PZQ group and by 66.04% in ILP-PQZ group of mice (Fig 12B).

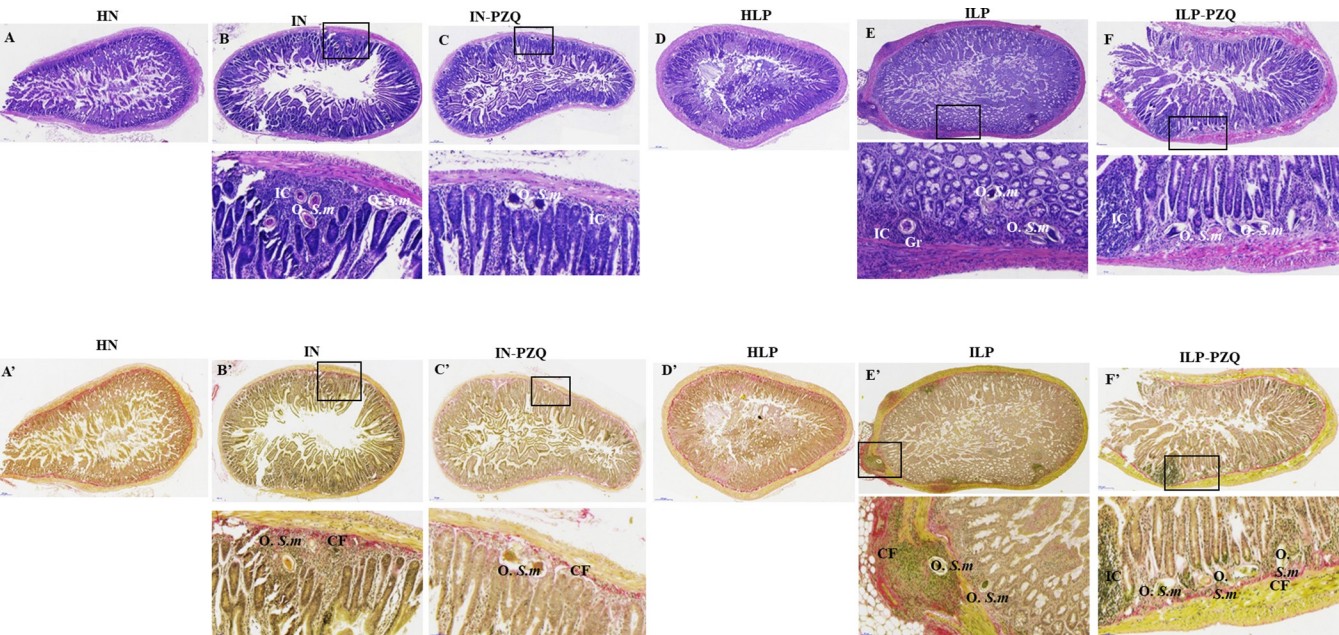

**Fig 11. Effects of praziquantel treatment on the intestine histology of *Schistosoma mansoni*-infected mice receiving a low-protein diet.** A, B, C, D E, and F: Hematoxylin-eosin staining A', B', C', D', E' and F': Picrosirius staining Group IN: infected-untreated mice receiving a standard diet; group IN-PZQ: infected mice receiving a standard diet and treated with praziquantel; group ILP: infected-untreated mice receiving a low-protein diet; group ILP-PZQ: infected mice receiving a low-protein diet and treated with praziquantel. IC: inflammatory cells, Gr: granuloma, O. S.m: egg of *S. mansoni*, CF: collagen fiber.

The volume of the liver granulomas significantly decreased by 38.37% in IN-PZQ mice compared to that of the control IN group. In the group of infected mice receiving the low-protein diet and treated with praziquantel, the hepatic granulomas volume was close to the one of the infected untreated mice group. (Fig 12C). Praziquantel treatment did not correct the hepatic granulomas volume in the ILP-PZQ group of mice.

## Discussion

Protein malnutrition and schistosomiasis are health problems that affect millions of people. A low-protein diet increases the risk of illness and death [7,8]. Protein malnutrition can impair immune function and affect hematopoiesis, biochemical and histological parameters [7,10,46–50]. Consequently, a low-protein diet increases susceptibility to infections and induces an imbalance between food intake and the need to ensure the most favourable growth [47,51–53]. As revealed by the Lee index, the nutritional state of healthy mice receiving the low-protein diet was normal in the current study. Since the bromatological characteristics of the low-protein diet and the standard one revealed that the total metabolizable energy was similar, it can be understandable that the low-protein diet used to feed healthy mice has supplied food intake without altering energy expenditure, body fat, lean mass, and body weight [54,55]. While infected with *S. mansoni*, mice receiving either the standard or the low-protein diet significantly lost weight, probably due to anaemia and hypoglycemia. Indeed, the current study clearly showed a reduction in red blood cell count, hematocrit and glucose concentration in infected mice as *S. mansoni* adult worms use haemoglobin and glucose for their nutrition, energy supply, and egg-laying [56]. As determined by anthropometric indexes, growth retardation has been associated with chronic *S. mansoni* infection in children [17]. The Lee index also

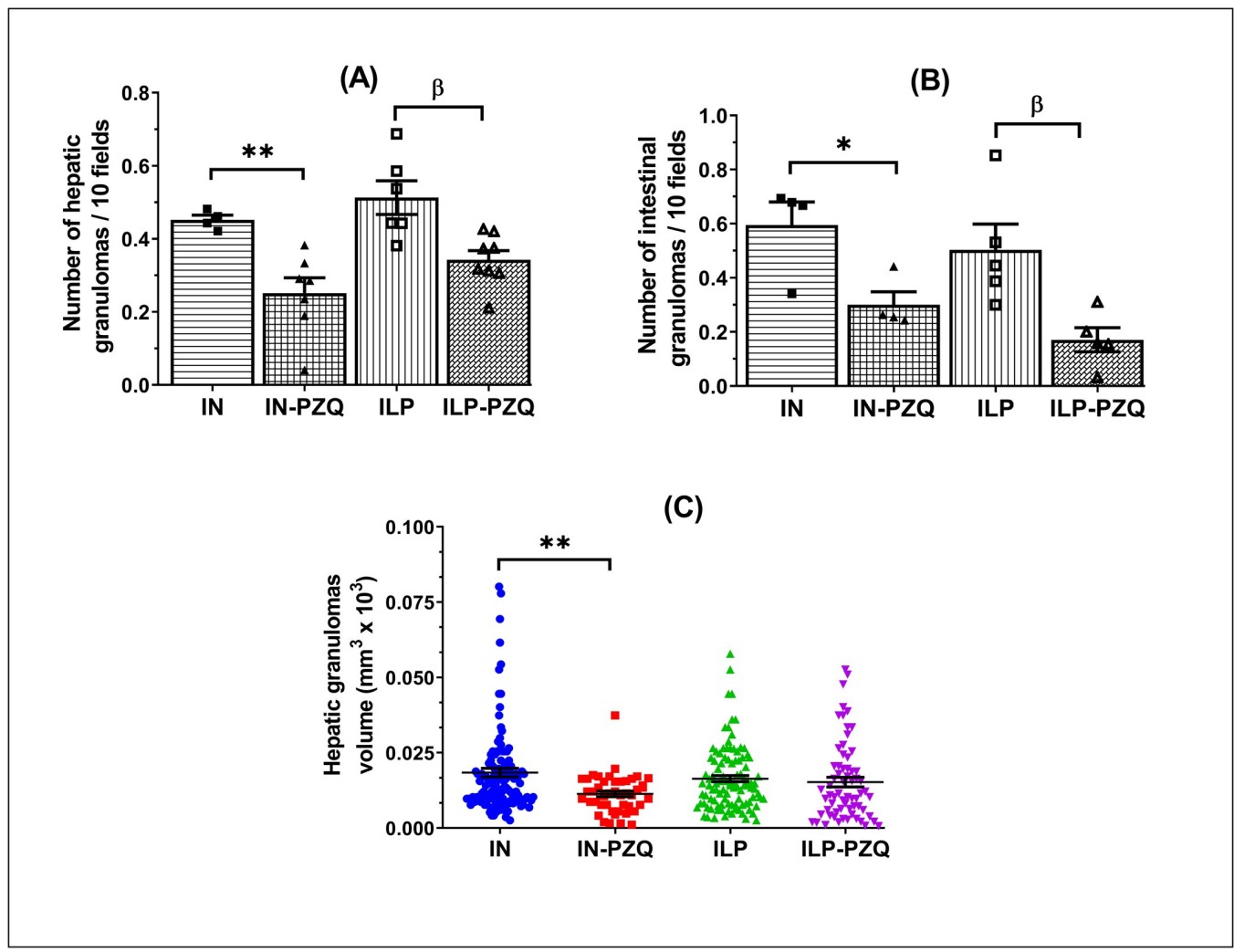

**Fig 12. Effects of praziquantel treatment on the liver and intestine granulomas number and volume of *Schistosoma mansoni*-infected mice receiving a low-protein diet.** Data are expressed as mean ±SEM (n = 4–8). Group IN: infected-untreated mice receiving a standard diet; group IN-PZQ: infected mice receiving a standard diet and treated with praziquantel; group ILP: infected-untreated mice receiving a low-protein diet; group ILP-PZQ: infected mice receiving a low-protein diet and treated with praziquantel. ANOVA followed by Turkey's multiple comparison test. $^*P < 0.05$; $^{**}P < 0.01$: values are significantly different from infected-untreated mice receiving the standard diet (group IN). $^{\beta}P < 0.05$; values are significantly different from infected-untreated mice receiving the low-protein diet (group ILP).

provides information on the health and growth of individuals and is often cited as a reliable indicator of nutritional status [57,58]. In this study, the Lee index of infected mice receiving the low-protein diet was significantly reduced, demonstrating the negative impact of undernutrition and schistosomiasis on growth [21,59]. Moreover, the total proteins, albumin, and glucose concentrations of mice fed with the low-protein diet significantly decreased. Hypoproteinemia and hypoalbuminemia could be the consequence of reducing protein supply in the diet, leading to a decrease of protein biosynthesis by the host. The combined effects of protein and glucose deficiencies and *S. mansoni* infection have undoubtedly impaired the nutritional status of the mice and, therefore, their growth. Contrary to that of infected mice fed a low-protein diet, oral administration of PZQ improves the body weight of infected mice receiving the standard diet. The low body weight of the infected mice fed with the low-protein

diet and treated with PZQ could be linked to the persistence of anaemia and protein and glucose deficiencies after PZQ treatment. The red blood cell counts, and the concentrations of total proteins, albumin, and glucose of these mice remained low compared to those of healthy mice. An adverse synergistic action of malnutrition and *S. mansoni* infection on host glycemia and hemoglobinemia have been previously described [23,24].

In the current study, the worm recovery rates of infected-untreated mice were similar in both diet groups, indicating the capacity of worms to accomplish their life cycle in a final host fed with a low-protein diet [18,60]. However, some authors have mentioned that male adult *S. mansoni* isolated from mice fed with a low-protein diet present thinning of the tegument, large tubercles on the dorsal region, vacuolated areas on the subtegumental region, and small testes [18]. On the other hand, Wolowczuk et al. [61] pointed out that even if a low-protein diet may induce morphological changes in schistosomes and produce dwarf worms, these changes would not be enough to impair their reproductive capacity; dwarf worms would still be able to mate and lay eggs. This correlates with our study, where egg loads in feces and liver were comparable in both diet groups. *S. mansoni*-infected mice also exhibited hepatosplenomegaly and intestine enlargement, undoubtedly due to egg deposition in the hepatic and intestine parenchyma. Enlargement of the spleen could result from passive congestion of blood flow and reticuloendothelial hyperplasia [60,62]. Oral administration of PZQ to infected mice receiving the standard diet (IN-PZQ group) resulted in a significant reduction of hepatomegaly, splenomegaly, and intestine enlargement. This could be the consequence of the considerable decrease of egg burden in the liver and intestine after PZQ treatment, as recorded in this study and by some authors [30,63–65]. The reduction of hepatic egg load of infected mice fed with the low-protein diet and treated with PZQ (ILP-PZQ) was insufficient to induce a reduction of hepatomegaly. Indeed, despite PZQ treatments, the worm and hepatic egg burdens of infected mice receiving the low-protein diet were higher than those of infected mice receiving the standard diet. These findings justify the persistence of anemia and low glucose levels in these mice and indicate that a low-protein diet could reduce PZQ efficacy. This is probably linked to intestinal malabsorption of PZQ as França et al. [13] have demonstrated that in undernourished mice, intestinal villi lose their brush border and become small and irregular, thus impairing intestinal function. Ibrahim et al. [12] pointed out that intestinal inflammation resulting from enteric pathogens disrupts intestinal barrier function. Undernutrition coupled with granulomatous intestinal inflammation, as recorded in infected mice fed with a low-protein diet, could thus explain the malabsorption of PZQ. The poor efficacy of PZQ in protein-undernourished and infected mice could also be understandable by exploring the PZQ mechanism of action. PZQ acts on schistosome motility by disrupting calcium ions homeostasis in the worm. It alters the worm's membrane permeability, causing an uncontrolled and rapid calcium ion influx that leads to sustained muscle contraction and paralysis [66–68]. It has been demonstrated that PZQ-induced disruption of schistosome tegument and muscular contraction depends on calcium concentration in the worm's environment [67,69–72]. In the current study, calcium levels remained low in infected mice receiving the low-protein diet and treated with PZQ. The extracellular calcium could therefore be insufficient to sustain schistosome muscle contractions which leads to their paralysis and death. This might also explain the important worm and egg burdens of infected mice receiving the low-protein diet after PZQ treatment.

Embolization of schistosome eggs in the liver induces impairment of hepatic metabolism. Following da Silva et al. [73], who demonstrated lipid metabolism alterations in hepatosplenic schistosomiasis, the present study showed significant decreases in total cholesterol, cholesterol LDL, and triglyceride in infected untreated mice receiving the standard or the low-protein diet. Because *S. mansoni* worms cannot synthesize cholesterol required for their growth and egg production, they absorb it from the host's bloodstream, hence dropping total cholesterol,

cholesterol LDL, and triglycerides [74]. Hepatocellular injury leading to the modification of transaminases (ALT and AST), alkaline phosphatase (ALP), gamma-glutamyl transferase (GGT), and total bilirubin (BIL) levels is also common in *S. mansoni* infection [30,45,63,65]. The presence of *S. mansoni* eggs in the liver parenchyma initiates an inflammatory process that gives rise to the formation of granulomas. These granulomatous lesions injure hepatocytes, which lose their membrane integrity and release transaminases in the bloodstream, thus increasing their activity [60,75]. The present investigation also showed increased PAL, GGT, and BIL concentration, implying an impairment of hepato-biliary function [73]. Since the liver is the site of iron metabolism, hepatocyte injury could increase iron concentration as recorded in *S. mansoni*-infected mice during this study and in patients with chronic liver diseases [76]. Administration of PZQ to infected mice receiving the standard diet or the low-protein diet ameliorates the lipid profile, ALP activity, and BIL concentration. ALT and GGT activities and iron levels were improved only in infected mice receiving the standard diet. These results demonstrated that the liver is recovering from the injury induced by *S. mansoni* infection. The concomitant reduction of worm burden and egg load in the liver after PZQ treatment has contributed to reducing the hepatocellular damage and probably initiated liver regeneration. Oliva-Vilarnau et al. [77] have demonstrated that calcium is a critical component of hepatic growth factors signalling during liver regeneration after injury. Intracellular calcium has been associated with mitogens epithelial growth factor (EGF) and hepatocyte growth factor (HGF) in hepatocytes. As recorded in this study, the normalization of the calcium concentration of infected mice receiving the standard diet after PZQ treatment could be essential for their liver regeneration. Remarkably, in *S. mansoni* infected mice receiving the low-protein diet, the ALT and GGT activities and iron and calcium concentrations did not recover after PZQ treatment. This impaired recovery is probably due to the high liver egg burden and hypocalcemia in these mice, indicating a diminished efficiency of PZQ treatment in this low-protein setting again.

The infection induced liver oxidative stress in the current study, marked by malondialdehyde overload and depletion of antioxidants and nitrites. During schistosomiasis, the granuloma-inflammatory cells generate reactive oxygen species (ROS) such as superoxide and hydroxyl radicals involved in the production of lipid peroxides. This leads to an increased concentration of MDA in the liver. Because of its implication in the generation of peroxynitrite, the oxidization of nitric oxide (NO) to nitrites, and consequently the concentration of nitrites, diminish. Therefore, the host will immediately use antioxidants to counteract the harmful action of ROS overload. The consequence will be the reduction of hepatic SOD, CAT, and GSH levels [4,30,45,63–65,78,79]. Administration of PZQ to *S. mansoni*-infected mice fed with a standard or a low-protein diet reduced lipid peroxidation (MDA) and improved enzymatic, non-enzymatic antioxidants levels (SOD, CAT, and GSH). However, despite PZQ treatment to infected mice fed with the low-protein diet (ILP-PZQ), MDA concentration was still higher, and nitrites were lower than those of infected mice fed with the standard diet treated with PZQ (IN-PZQ). It implies that despite the treatment, lipoperoxidation due to ROS and peroxynitrite production occurs, probably because of the high liver egg burden of the ILP-PZQ group of mice. Indeed, the action of PZQ on oxidative stress is indirect through its schistosomicidal effect on adult worms. By limiting the recruitment of inflammatory cells, the source of ROS, in the vicinity of schistosome eggs, PZQ indirectly reduces ROS production [64,80,81].

The generation of oxidative stress is intimately linked to the granulomatous inflammatory cells' activity. Therefore, reducing ROS production would reduce inflammatory response or vice versa. We assessed the effects of PZQ treatment on the inflammatory status of *S. mansoni*-infected mice by performing histomorphometry of the liver and intestine and determining the immunological status of the mice. Histopathological examination of the liver and the mall intestine sections of infected mice fed with the standard or the low-protein diet and treated

with PZQ (IN-PZQ and ILP-PZQ, respectively) revealed fewer and smaller granulomas than in infected untreated mice. These observations were confirmed by the reducing number of liver and intestine granulomas. The hepatic granulomas volume also decreased in the IN-PZQ group of mice but not in the ILP-PZQ group. Other authors obtained similar results on *S. mansoni*-infected mice receiving a standard diet. They correlated this anti-inflammatory activity of PZQ to its schistosomicidal effect resulting in the reduction of eggs laying in the liver and intestine [30, 45, 82–84]. The decrease in granulomas volume after PZQ treatment of infected mice could also be consistent with the reduction of fibrosis. This is materialized by the decline of collagen types I and III or their biomarker hydroxyproline [64, 82, 84]. The non-reduction of the hepatic granulomas volume of infected mice receiving the low-protein diet and treated with PZQ could be explained by the continuous recruitment of inflammatory cells due to the critical egg burden in their liver. It has been demonstrated that schistosomiasis and undernutrition comorbidity led to increased egg production and liver damage marked by high density and large areas of exudative granulomas [19, 20].

Malnutrition considerably impairs the immune system by causing atrophy of primary lymphoid organs, compromising complement components and phagocyte function, and decreasing the biological function of lymphocytes, macrophages, and Kupffer cells [12,13]. In the current study, protein deficiency did not induce significant Th1, Th2, Th17, and Treg levels. In contrast, several authors have reported reduced proliferation and effector function of lymphocytes T, a low level of blood lymphocytes T, and a shift of a Th1 towards a Th2 cytokine response in children with severe malnutrition and fasted mice or mice fed with a low-protein diet [13–15,85–88]. The lack of variation in cytokine levels in our study could be due to the moderate low-protein diet used to feed mice (14.60% of protein), as compared to the very low-protein diet (2% of protein) used by others [89]. The involvement of T lymphocytes, especially CD4$^+$ T cells, in the immune response against schistosomiasis is essential [90]. During the migratory phase of schistosomula, from 3 to 5 weeks post-infection, the dominant immune response is Th1. When parasites mature, mate and begin to lay eggs at 5 to 6 weeks post-infection, female worms release fertilized eggs that stimulate a Th2 immune response via their soluble egg antigens. The Th2 response reaches a peak at approximately 8 weeks post-infection and is down-modulated with progression to chronic infection [91–94]. In the current study, the immune response at 9 weeks of *S. mansoni* infection shifted from a Th1 to a Th2 response in infected mice receiving the standard or the low-protein diet. Indeed, the serum levels of TNF-α and IFN-γ, as well as the liver mRNA expression of CCL2/MCP-1, FGF, CCL3/MIP 1-α, CXCL-10/IP-10, and IFN-γ decreased significantly. Concurrently, the serum levels of IL-4, IL-5, IL-13, and the liver mRNA expression of FoxP3 increased. During the early acute stage of *S. mansoni* infection, a Th1 response is initiated and is characterized by the increased concentration of inflammatory cytokines and chemokines [95]. The elevated expression of eosinophil-, neutrophil- and macrophage- associated chemokines such as CCL2/ MCP-1, CCL3/MIP 1-α, and CXCL-10/IP-10 is concomitant to the migration of eosinophils and neutrophils from the circulation to the site of the granulomatous inflammation, and the recruitment and activation of hepatic stellate cells (HSCs). [96–98]. HSCs can also be activated during the chronic stage of schistosomiasis, and in turn, they can produce chemokines like CCL2, CCL3, and CXCL-10 following liver injury [99]. The reduction of the mRNA expression of CCL2, CCL3, FGF, CXCL-10, and IFN-γ at the 9$^{th}$-week post-infection reflects a downmodulation of the Th1 response by reducing the recruitment of inflammatory cells and HSCs to the granulomatous site. This indicates a maximal granuloma growth, as revealed by the highest IL-2 production [100]. The increased serum level of IL-17A during this period is also correlated to the granulomatous lesions associated with severe liver pathology [95,101]. Hepatic granuloma formation and fibrosis are upregulated by Th2 and Th17 cells, mainly secreting IL-4 and IL-17A,

respectively [102–104], and downregulated by Th1 and Treg cells [105,106]. Regulatory T cells secrete IL-10 and TGF-β that suppress the activation of dendritic cells, mediate Th1 and Th2 responses and inhibit granuloma development and fibrosis during *S. mansoni* infection to promote host survival [90,95,107,108]. During this study, increasing the serum level of IL-10 and its mRNA expression in infected mice demonstrates its regulatory activity by inhibiting the production of cytokines Th1 like TNF-α [107]. IL-10 can also play its regulatory role in periportal fibrosis by blocking the activation of quiescent HSCs [109,110]. The reduction of the mRNA expression of TGF-β1 in *S. mansoni*-infected mice after 9 weeks could also reflect the modulation of liver fibrosis. It has been reported that high levels of TGF-β1 are associated with liver fibrosis and pulmonary arterial hypertension in *S. mansoni* infection [111,112]. The normalization of Th1, Th2, Th17, and Treg cytokine levels (IL-2, IL-5, IL-13, IL-17A, and TGF-β1) and the mRNA expression of some chemokines and Treg cytokines (CCL3, CXCL-10 and TGF-β1) after PZQ treatment is the consequence of drug-induced clearance of *S. mansoni* worms. This reduces the number of eggs trapped in the liver, and subsequently the recruitment and migration of inflammatory cells around the eggs. Indeed, in the current study, PZQ treatment re-establish normal levels of total leukocytes and eosinophils that were significantly increased and of lymphocytes that was decreased by *S. mansoni* infection. Leukocytosis, eosinophilia and lymphocytopenia are common during granulomatous inflammatory diseases as schistosomiasis. Blood total leukocytes and eosinophils increase to fight the infection and lymphocytes migrate at the sites of inflammation, being attracted by chemokines and cytokines. This migratory process leads to a decrease of their blood count [113,114]. The capacity of PZQ to normalize the total leukocyte, eosinophil and lymphocyte blood counts therefore demonstrates its ability to limit the immunogenic action of *S. mansoni* eggs and to alleviate the infection. Because of the persistent high worm and egg burdens in the group of infected mice receiving the low-protein diet and treated with PZQ, the levels of IL-5 and the mRNA expression of CCL3 and CXCL-10 were not ameliorated by the treatment. FoxP3 was barely detectable in non-infected mice, but expressed in the liver of infected untreated mice and of infected mice treated with PZQ. This is consistent with its down-regulatory role on Th1 and Th2 cytokines production and on the fibrogranulomatous inflammation via the inhibition of the profibrogenic activity of IL-4 and IL-13 [115,116].

## Conclusion

The current study revealed that a low-protein diet resulted in the low efficacy of praziquantel treatment on the growth, hepatomegaly, worm burden, and egg output of *S. mansoni*-infected mice. The difference was also noted for liver function biomarkers such as ALT, GGT, iron, and calcium and the liver oxidative stress biomarker malondialdehyde. Furthermore, the immunomodulatory activity of praziquantel was not significant in these mice, as the levels of Th2 cytokine IL-5 and the mRNA expression of chemokines CCL3/MIP-1α and CXCL-10/IP-10 were not improved. This poor praziquantel efficiency was also observed on the liver granulomas volume. This study, therefore, demonstrated the reduced praziquantel efficacy in *S. mansoni*-infected mice receiving a low-protein diet. It also underlines the importance of targeting protein deficiency and malnutrition in populations living in schistosomiasis endemic areas for efficient disease control.

## Declarations

### Consent for publication

Not applicable.

## Acknowledgments

The authors are grateful to the association "Pathologie, Cytologie et Développement" (PCD), which kindly donated equipment and reagents for histological study. We also thank Ms Laetitia Béal and Ms Chloé Gommenginger for their technical assistance in realizing multiplex assays.

## Author Contributions

**Conceptualization:** Joseph Bertin Kadji Fassi, Hermine Boukeng Jatsa, Catherine Cannet, Alexander Wilhelm Pfaff, René Kamgang, Pierre Kamtchouing, Louis-Albert Tchuem Tchuenté.

**Data curation:** Joseph Bertin Kadji Fassi, Hermine Boukeng Jatsa.

**Formal analysis:** Joseph Bertin Kadji Fassi, Hermine Boukeng Jatsa, Ulrich Membe Femoe, Valentin Greigert.

**Funding acquisition:** Ulrich Membe Femoe, Ahmed Abou-Bacar.

**Investigation:** Joseph Bertin Kadji Fassi, Ulrich Membe Femoe, Valentin Greigert, Julie Brunet, Catherine Cannet, Christian Mérimé Kenfack, Nestor Gipwe Feussom, Emilienne Tienga Nkondo, Ahmed Abou-Bacar.

**Methodology:** Joseph Bertin Kadji Fassi, Hermine Boukeng Jatsa, Ulrich Membe Femoe, Valentin Greigert, Catherine Cannet, Ahmed Abou-Bacar, Alexander Wilhelm Pfaff, Louis-Albert Tchuem Tchuenté.

**Project administration:** Hermine Boukeng Jatsa, Alexander Wilhelm Pfaff.

**Supervision:** Hermine Boukeng Jatsa, Alexander Wilhelm Pfaff, René Kamgang, Pierre Kamtchouing, Louis-Albert Tchuem Tchuenté.

**Validation:** Hermine Boukeng Jatsa, Alexander Wilhelm Pfaff, Pierre Kamtchouing, Louis-Albert Tchuem Tchuenté.

**Visualization:** Joseph Bertin Kadji Fassi, Hermine Boukeng Jatsa, Ulrich Membe Femoe.

**Writing – original draft:** Joseph Bertin Kadji Fassi, Hermine Boukeng Jatsa.

**Writing – review & editing:** Hermine Boukeng Jatsa, Valentin Greigert, Alexander Wilhelm Pfaff, Louis-Albert Tchuem Tchuenté.

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
