## [Decision Letter · Decision Letter 0]

11 Apr 2022

Dear Pr Boukeng Jatsa,

Thank you very much for submitting your manuscript "Protein undernutrition reduces the efficacy of praziquantel in a murine model of Schistosoma mansoni infection" for consideration at PLOS Neglected Tropical Diseases. As with all papers reviewed by the journal, your manuscript was reviewed by members of the editorial board and by several independent reviewers. In light of the reviews (below this email), we would like to invite the resubmission of a significantly-revised version that takes into account the reviewers' comments. 

Dear Dr. Fassi and colleagues,

In general, the reviewers were positive with regards to your submission. The research question of the effects of malnutrition on the efficacy of praziquantel has important clinical and biological implications, and the research approach taken was sound.

Please address the critiques of the reviewers. In particular, both reviewers highlighted that the presentation of the results in some areas was confusing and made recommendations regarding changing many of the tables into graphs. Additionally, the reviewers recommended highlighting several limitations to the study in the discussion.

We cannot make any decision about publication until we have seen the revised manuscript and your response to the reviewers' comments. Your revised manuscript is also likely to be sent to reviewers for further evaluation.

Sincerely,

Edward Mitre

Associate Editor

Sergio Oliveira

Deputy Editor

Dear Dr. Fassi and colleagues,

In general, the reviewers were positive with regards to your submission. The research question of the effects of malnutrition on the efficacy of praziquantel has important clinical and biological implications, and the research approach taken was sound.

Please address the critiques of the reviewers. In particular, both reviewers highlighted that the presentation of the results in some areas was confusing and made recommendations regarding changing many of the tables into graphs. Additionally, the reviewers recommended highlighting several limitations to the study in the discussion.

Reviewer's Responses to Questions

**Key Review Criteria Required for Acceptance?**

**Methods**

-Are the objectives of the study clearly articulated with a clear testable hypothesis stated?

-Is the study design appropriate to address the stated objectives?

-Is the population clearly described and appropriate for the hypothesis being tested?

-Is the sample size sufficient to ensure adequate power to address the hypothesis being tested?

-Were correct statistical analysis used to support conclusions?

-Are there concerns about ethical or regulatory requirements being met?

Reviewer #1: see below

Reviewer #2: See general comments

**Results**

-Does the analysis presented match the analysis plan?

-Are the results clearly and completely presented?

-Are the figures (Tables, Images) of sufficient quality for clarity?

Reviewer #1: see below

Reviewer #2: See general comments

**Conclusions**

-Are the conclusions supported by the data presented?

-Are the limitations of analysis clearly described?

-Do the authors discuss how these data can be helpful to advance our understanding of the topic under study?

-Is public health relevance addressed?

Reviewer #1: see below

Reviewer #2: See general comments

**Editorial and Data Presentation Modifications?**

Reviewer #1: see below

Reviewer #2: See general comments

**Summary and General Comments**

Reviewer #1: Fassi et al have studied the effect of protein undernutrition on the efficacy of praziquantel treatment on S mansoni infection. The authors find that PZQ efficacy is reduced as consequence of protein undernutrition in experimental model of murine schistosomaisis. 

This is a highly relevant research question populations in which S mansoni infection is endemic, often also suffer from malnutrition, including insufficient protein intake. Knowing how PZQ efficacy is affected by host nutritional status is therefore highly relevant. Alhtough the experimental approach is generally sound, the way the manuscript is written and data are presented make it hard to read. This should be improved. Specifically, I have the following comments:

Major comments:

1) The results section lists the main differences observed between the different experimental groups. However, it is now very difficult to read because it contains no rationale why certain parameters are analyzed. In addition, no summarizing sentences present at the end of each section, to bring across the main points of each of the findings. Also the headings of each section should reflect the findings of that section rather than what is measured. So, for instance, not ‘worm burden and egg load….in mice with a protein deficient diet and treated with PZQ’ but ‘PZQ treatment is less efficacious in reducing worm burden etc in Sm infected mice on a protein-poor diet’ These things should be changed, to improved readability. 

2) The results section starts off with 10 tables, which are much harder to read then bar graphs. So please change them into bar graphs and display individual mice as dots, to be able to gauge inter experimental variation. 

3) What was the rationale to treat the mice with PZQ at day 36 post infection, which is the prepatent phase, before egg deposition? In clinical settings, treatment will most likely take place in infections where egg laying has already started.

4) Where the mice randomized based on weight before infection? It doesn’t seem so, as there were some weight difference between the groups before Sm infection (table 4). This may have skewed some of the results

5) The authors attribute difference in PZQ treatment efficacy to differences in protein intake. However, before Sm infection, there seems to be a weight difference between the two groups with different diets. Hence differences in efficacy may inf fact be attricbuted to initial weight differences, rather than differences in protein intake specifically. This could be have solved by performing paired feeding experiments, where caloric intake of the mice on a protein poor diet was matched by those on a standard diet. This should at least be discussed and mentioned as limitation of the study. 

6) Please also measure type 2 cytokines in figure 3.

7) What about hydroxyproline levels in livers of these mice as a measure for fibrosis? 

Minor comments:

1) The authors use the term ‘protein-deficient diet’. However the type of diet used in this study is reduced in protein content, not deficient. This should be corrected throughout the manuscript

2) Please explain in the results section what the Lee index entails. This is explained later on, but should be explained at time of introduction

3) Abbreviations of the experimental groups don’t have to be introduced in every single results section. Once at the beginning of the results section is enough

Reviewer #2: Inadequate feeding is common in areas of poor sanitation where diseases such as schistosomiasis prevail. Kwashiorkor, a severe form of malnutrition common in some of these countries argue for the clear lack of protein-sufficient diet in these countries.

 In this report, Jatsa and collaborators assessed the impact of protein-deficient diet on the efficacy of anti-schistosomiasis treatment with Praziquantel.

 Using a mouse model of infection, the authors fed young mice (<5 weeks old) with protein-sufficient or protein-deficient diets for 5-6 weeks and infected them individually with 50 S mansoni cercariae. Animals were treated for 5 consecutive days with 100 mg/kg of PZQ after 6 weeks of infection and then culled after 9 weeks of infection to retrieve parasitological, pathological, biochemical and immunological markers of infection.

The authors report on a reduced ability of PZQ to clear parasites, restore ion levels, improve pathology and ameliorate stress in S mansoni-infected and PZQ treated protein-deficient diet fed mice where as PZQ treatment in S mansoni infected mice and initially fed with protein-sufficient diet displayed reduced parasitological burden, ameliorated liver pathology and reverse immunomodulatory markers that are pathognomonic of s mansoni infection.

The authors went on to conclude on the possible negative influence of a protein-deficient diet on the schistosomicidal, antioxidant, anti-inflammatory and immunomodulatory activities of PZQ.

General Comments:

The report is conceptually novel in its approach and does attempt to tackle a clearly relevant facet of schistosomiasis control in poor areas where undernutrition is common. As such the potential in providing preclinical insights on the current impact of undernutrition on PZQ treatment efficiency during schistosomiasis is clear and a novel addition to the current body of literature. The massive amount of data and the comprehensive appraisal of the differences in the experimental set-ups do considerably strengthen the robustness of the present report.

However, some points of concern are to be raised here to provide more clarity to the conclusions made by the authors and maximize the impact of the present report:

1) The present report of body weight change in table 4 is quite cryptic and should be rather displayed in a graph (XY, linear regression option on graph pad). Having performed the same with the data provided by the authors in table 4, I did fail to clearly see the claimed difference. In fact, weight loss was not to be found for all infected animals as claimed by the authors, especially for PZQ treated groups at day 106 i.e. prior to treatment per se. The data do not support a reversal of weight loss but perhaps a prevention of weight loss, as presented. Perhaps a body weight variation curve (i.e. following up the changes of an animal body weight and plotting that over time) would be more informative. As it now stands, the clear baseline differences in body weight do preclude a conclusive analysis of the extent of body weight change per group.

2) Lines 109-112:’ All procedures in this study followed the principles of laboratory animal use and care of the “European Community” guidelines (EEC Directive 2010/63/EEC) and were approved by the “Animal Ethical Committee” of the Laboratory of Animal Physiology of the Faculty of Sciences, University of Yaoundé I – Cameroon.’ The authors should provide an Ethics approval number for the study as is commonly assigned by established animal ethics committee. 

3) Similarly to the body weight in table 4, the organ weight indexes in table 5 would benefit from a bar graph representation of the different indexes coupled with a conclusive curve on the percentage change from naïve to infected and from infected to treated. It is paramount here to consider the reference to the naïve setting towards which the PZQ treatment should move the system rather than simply claiming differences induced on the infected by the PZQ treatment (IN-PZQ vs IN compared to ILP-PZQ vs ILP). The latter approach, taken by the authors, is misleading as it is clear that infection does not similarly affect the IN and the ILP mice when compared to their naïve controls HN and HLP respectively. 

4) Table 6 should also be displayed as bar graph for clarity. It is also necessary here to plot a variation curve to show how different the efficacy of PZQ is in one setting over the other i.e. a bar graph of worm burden changes after PZQ treatment in each group. This is the most critical result here, and should be more clearly represented to convey the key message of the paper.

5) Table 7-10 should be modified into graphs also i.e. presenting the data as bar graph and then performing statistical comparisons between HN vs HLP (what the diet does); HN vs IN & HLP vs ILP (what the infections does in the groups and plotting the changes here from infection in N vs infection in LP); IN-PZQ vs IN and vs HN compared to ILP-PZQ vs ILP and vs HLP (what treatment does and how close back to naïve does it bring the system) etc.. PS: Note that table 10 is too large and truncated as it was inserted in the wrong orientation.

6) Figure 6: How do the authors explain such an observation; secondly, hepatic granuloma volumes are apparently reduced by PZQ treatment in IN-PZQ but not in ILP-PZQ whereas both do present a drastic abrogation of IL-13 production (figure 2), a key cytokine, along with IL-4, in granuloma formation. How could that be reconciled ?

7) Lines 736-737: The authors refer to an eventual decrease of fibrosis in their assays following PZQ treatment. Was this decline quantitatively reported in the figures (area coverage in liver sections or hydroxyproline per trapped eggs in tissue etc…) ? Also, was this fibrosis development affected in ILP when compared to IN mice ?

8) Does protein undernutrition affect the course of the disease + host responsiveness to the disease and/or does it affect the efficacy of PZQ? The protein undernutrition could very well impair the host immune system, a key actor in the killing process of worms by PZQ. In fact, after worm paralysis induced by the drug, the host immune system might play a critical role in finalizing the schistosomicidal activity of the treatment (Modha J, Lambertucci JR, Doenhoff MJ, McLaren DJ: Immune dependence of schistosomicidal chemotherapy: an ultrastructural study of Schistosoma mansoni adult worms exposed to praziquantel and immune serum in vivo. Parasite Immunol. 1990, 12 (3): 321-334. 10.1111/j.1365-3024.1990.tb00958.x.). This should be better discussed and the a more inclusive view incorporating this aspect should be reflected in the entire discussion.

PLOS authors have the option to publish the peer review history of their article (what does this mean?). If published, this will include your full peer review and any attached files.

Reviewer #1: No

Reviewer #2: Yes: Justin Komguep Nono
---

## [Editor Report · Decision Letter 1]

28 Jun 2022

Dear Pr Boukeng Jatsa,

We are pleased to inform you that your manuscript 'Protein undernutrition reduces the efficacy of praziquantel in a murine model of Schistosoma mansoni infection' has been provisionally accepted for publication in PLOS Neglected Tropical Diseases.

Best regards,

Edward Mitre

Associate Editor

Sergio Oliveira

Deputy Editor

---

## [Editor Report · Acceptance letter]

8 Jul 2022

Dear Pr Boukeng Jatsa,

We are delighted to inform you that your manuscript, "Protein undernutrition reduces the efficacy of praziquantel in a murine model of Schistosoma mansoni infection," has been formally accepted for publication in PLOS Neglected Tropical Diseases.

Best regards,

Shaden Kamhawi

co-Editor-in-Chief

Paul Brindley

co-Editor-in-Chief
